# c-Myc uses Cul4b to preserve genome integrity and promote antiviral CD8⁺ T cell immunity

**Asif A. Dar** [1] ✉, **Dale D. Kim** [1], **Scott M. Gordon** [2], **Kathleen Klinzing** [1], **Siera Rosen**[1], **Ipsita Guha**[1], **Nadia Porter**[1], **Yohaniz Ortega**[1], **Katherine S. Forsyth** [1], **Jennifer Roof** [3], **Hossein Fazelinia**[3], **Lynn A. Spruce** [3], **Laurence C. Eisenlohr** [1,4], **Edward M. Behrens**[5] & **Paula M. Oliver** [1,4] ✉

During infection, virus-specific CD8⁺ T cells undergo rapid bursts of proliferation and differentiate into effector cells that kill virus-infected cells and reduce viral load. This rapid clonal expansion can put T cells at significant risk for replication-induced DNA damage. Here, we find that c-Myc links CD8⁺ T cell expansion to DNA damage response pathways though the E3 ubiquitin ligase, Cullin 4b (Cul4b). Following activation, c-Myc increases the levels of Cul4b and other members of the Cullin RING Ligase 4 (CRL4) complex. Despite expressing c-Myc at high levels, Cul4b-deficient CD8⁺ T cells do not expand and clear the Armstrong strain of lymphocytic choriomeningitis virus (LCMV) in vivo. Cul4b-deficient CD8⁺ T cells accrue DNA damage and succumb to proliferative catastrophe early after antigen encounter. Mechanistically, Cul4b knockout induces an accumulation of p21 and Cyclin E2, resulting in replication stress. Our data show that c-Myc supports cell proliferation by maintaining genome stability via Cul4b, thereby directly coupling these two interdependent pathways. These data clarify how CD8⁺ T cells use c-Myc and Cul4b to sustain their potential for extraordinary population expansion, longevity and antiviral responses.

During viral infection, antigen-specific CD8⁺ T cells undergo a massive expansion from below the threshold of easy detection ( < ~1000 cells/mouse) to >10⁷ antigen-specific CD8⁺ T cells by day 8 after infection. This expansion generates sufficient numbers of effector CD8⁺ T cells to aid the resolution of infection and generation of antigen-specific memory cells[1,2]. Such a rapid expansion requires a short cell cycle of ~4 h[3] which is significantly shorter than the cycling rate of most cancer cells[4]. This shortened cell cycle comes with a significant risk for replication stress and DNA lesions that jeopardize the viability of antigen-specific T cells. For this reason, CD8⁺ T cells are thought to have

powerful DNA damage response (DDR) mechanisms to maintain genome integrity. Supporting this, DDR pathways have recently been shown to be crucial in the formation of optimal CD8⁺ T cell memory and for CD8⁺ T cells used in cancer immunotherapy[5,6]. However, the mechanisms regulating CD8⁺ T cell cycle and DDR pathways during the generation of effector and/or memory CD8⁺ T cells remain poorly defined.

c-Myc is a transcriptional regulator that integrates cell proliferation, cell growth, and metabolism through the activation of numerous target genes. In developing tumors, c-Myc helps reduce DNA

[1]Division of Protective Immunity, The Children's Hospital of Philadelphia, Philadelphia, PA, USA. [2]Division of Neonatology, The Children's Hospital of Philadelphia, Philadelphia, PA, USA. [3]Division of Cell Pathology, The Children's Hospital of Philadelphia, Philadelphia, PA, USA. [4]Department of Pathology, University of Pennsylvania, Philadelphia, PA, USA. [5]Division of Rheumatology, The Children's Hospital of Philadelphia, Philadelphia, PA, USA. ✉e-mail: dara@chop.edu; paulao@pennmedicine.upenn.edu

replication stress either by maintaining a balanced nucleotide pool or through the expression of cell cycle or DNA repair proteins[7,8]. However, c-Myc is also identified as the first endogenous source of replication stress in both primary and transformed cells[9,10]. While c-Myc is known to drive proliferation and metabolic reprogramming in both activated T cells[11] and tumor cells[12], whether and how it participates in protecting T cells from the ensuing DNA damage is not known. c-Myc was recently shown to function during the first division of T cells by segregating asymmetrically in daughter cells and inducing an effector fate via an interaction with Brg1[13,14]. Importantly, CD8[+] T cells divide ~15–20 times during an acute infection with lymphocytic choriomeningitis virus (LCMV)[15]. Thus, whether c-Myc functions only in early fate programing or whether it controls DDR over the course of this proliferative burst is not known.

Rapidly dividing cells, including activated T cells, have developed sophisticated mechanisms to sense DNA damage. A central regulator of the DDR is the tumor suppressor p53. This transcription factor is activated by damage-responsive kinases and controls the cellular response by inducing the expression of target genes including p21. Both primary and memory antigen-specific proliferative responses in T cells require downmodulation of p53[16]. Downregulation of p53 following T cell activation could be the reason why wild type and p53[-/-] T cells generated similar Ag-specific T cell responses[17]. In contrast, T lymphocytes from p21[-/-] mice exhibit a significant proliferative advantage over wild-type cells following prolonged stimulation[18,19]. Further, p21 overexpression results in a decreased effector/memory T cell expansion and autoimmune symptoms[20]. This demonstrates that in T cells, p21 can work independently of p53 to control T cell proliferation. However, the mechanism explaining how TCR signaling terminates p21 expression and consequently permits antigen-specific clonal expansion is not known.

Interestingly, p21 is regulated at both the transcriptional and post-transcriptional levels. c-Myc can repress p21 transcription in both p53-sufficient and -deficient cells[21,22]. In tumor cells, the ubiquitin ligase complex Cullin RING Ligase 4b (Cul4b) can target p21 protein for degradation[23]. Both c-Myc and Cul4b are regulated by TCR signaling[24,25]. Whether they regulate the turnover of p21 in CD8[+] T cell to limit replication stress and promote proliferation is not known. Cul4b is a member of Cullin-RING E3 ubiquitin ligase family and it shares 82% sequence similarity with Cullin RING Ligase 4a (Cul4a)[26]. Compared to Cul4a, Cul4b features a longer N-terminus, which contains an extra nuclear localization signal (NLS) which is believed to mediate its nuclear localization[27]. Both Cul4a and Cul4b promote cancer progression[28]. Cul4b loss in CD4[+] T cells induced genomic instability thereby decreasing CD4[+] T cell proliferation and survival[25]. T cells lacking c-Myc are unable to proliferate, rendering them ineffective as a model for studying how c-Myc regulates T cell proliferation and the DNA damage response. CD8[+] T cells are developmentally programmed to divide at a faster rate than CD4[+] T cells[15,29] but their dependence on Cul4b for survival, proliferation and differentiation is not known. Further, it is unclear what controls Cul4b expression in T cells and the mechanisms Cul4b uses to maintain genome integrity in highly proliferating T cells remains poorly understood.

Here, we assessed c-Myc-mediated cellular remodeling in activated CD8[+] T cells by comparing the proteomes of control (WT) and c-Myc deficient (KO) CD8[+] T cells. Consistent with prior reports, we found that c-Myc alters T cell protein translation, RNA rewiring and cell proliferation. Interestingly, we also found that loss of c-Myc correlated with an altered DNA damage response. Following T cell activation, c-Myc was found to be bound to the transcriptional start site of Cul4b, and c-Myc-deficiency decreased mRNA and protein levels of Cul4b as well as levels of other components of the CRL4 complex. Thus, we posited that c-Myc was increasing Cul4b levels to promote genome integrity. Supporting this, we found that Cul4b was required for the early expansion of CD8[+] T cells and mice lacking Cul4b in their

T cells were unable to clear viral infection. Specifically, Cul4b-deficiency resulted in a nearly complete collapse in effector and memory differentiation of CD8[+] T cells shortly after viral infection. This collapse was associated with a substantial increase in DNA damage. Cul4b-deficient CD8[+] T cells accumulated high levels of p21 and Cyclin E2, fueling replication stress and triggering genomic instability. High p21 levels were associated with a reduced turn-over of the replication licensing factors (RLFs) Cdt1 and Cdc6. The accumulation of Cyclin E2 has the ability to induce premature entry of CD8[+] T cells into the synthetic phase of cell cycle causing replication stress induced genomic instability and cell cycle block. Our data suggest that c-Myc promotes the expression of Cul4b to regulate the levels of p21 and Cyclin E2, inextricably linking proliferation of CD8[+] T cell with DNA damage response pathways. Thus, c-Myc and Cul4b collaborate to support effector and memory CD8[+] T cell fate and anti-viral immune responses.

## Results

### Following TCR signaling c-Myc enlists DNA damage response pathways and upregulates Cul4b in CD8[+] T cells

Recently we demonstrated that Cul4b protects CD4[+] T cells from replication induced DNA damage to allow CD4[+] T cell expansion[25]. In this study, we sought to determine whether c-Myc has a similar capacity for maintaining genomic integrity in CD8[+] T cells and, if so, whether Cul4b and c-Myc work collaboratively. As proof of principle, we did comparative analysis of biological processes regulated by c-Myc and Cul4b in CD4[+] T cells using Gene Ontology (GO) analysis. The GOplot algorithm was used to visualize the biological processes as depicted in the GObubble plot (Supplementary Fig. 1a, b). Cul4b predominantly regulated the biological processes linked to DNA damage response and cell cycle progression (Supplementary Fig. 1a), while c-Myc was annotated to be involved in many biological processes linked to protein translation, ribosome biogenesis, DNA damage response and cell cycle (Supplementary Fig. 1b). Notably, in both datasets we observed an enrichment of related or redundant GO terms (i.e., cell cycle, cell division, DNA repair and DNA damage response). Based on well described roles for c-Myc in transcription and translation and Cul4b in protein degradation, these data supported that c-Myc could drive the expression of cell cycle and DNA damage response pathways while as Cul4b would aid post-translational control.

How c-Myc and Cul4b cooperate in CD8[+] T cells to maintain genome stability and limit replication stress to allow clonal expansion is yet unknown. To assess this, we analyzed control and c-Myc[cKO] CD8[+] T cells using quantitative mass spectrometry. The control and c-Myc[cKO] CD8[+] T cells were stimulated for 24 h with anti-CD3/CD28 mAbs and then were subjected to mass spectrometry[30]. This duration of time induces an increase in cell size of the activated cells with no difference in survival between control (WT) and c-Myc[cKO] (KO) T cells. To quantify changes in protein expression, we compared protein abundances (copy number) in control and c-Myc-deficient CD8[+] T cells. Biological processes regulated by c-Myc in CD8[+] T cells following anti-CD3/CD28 activation were functionally annotated using the DAVID bioinformatics tool. The GOplot algorithm was used to visualize biological processes and are depicted in the GObubble plot (Fig. 1a). c-Myc was annotated to be involved in many biological processes; the top ten biological processes linked to translation, ribosome biogenesis and cell cycle progress and DNA damage response (Fig. 1a). The role of c-Myc, downstream of T cell activation, in translation and ribosome biogenesis has been well studied[30,31]; however, molecular mechanisms by which c-Myc controls cell cycle progression and the DNA damage response in T cells is not fully understood. We used GSEA to identify biological processes that are associated with the loss of c-Myc. The results revealed that biological processes linked to cell cycle, DNA repair, and cellular response to DNA damage were indeed enriched in control cells over c-Myc deficient CD8[+] T cells, as characterized by GSEA using the proteomic database (Fig. 1b, c and Supplementary

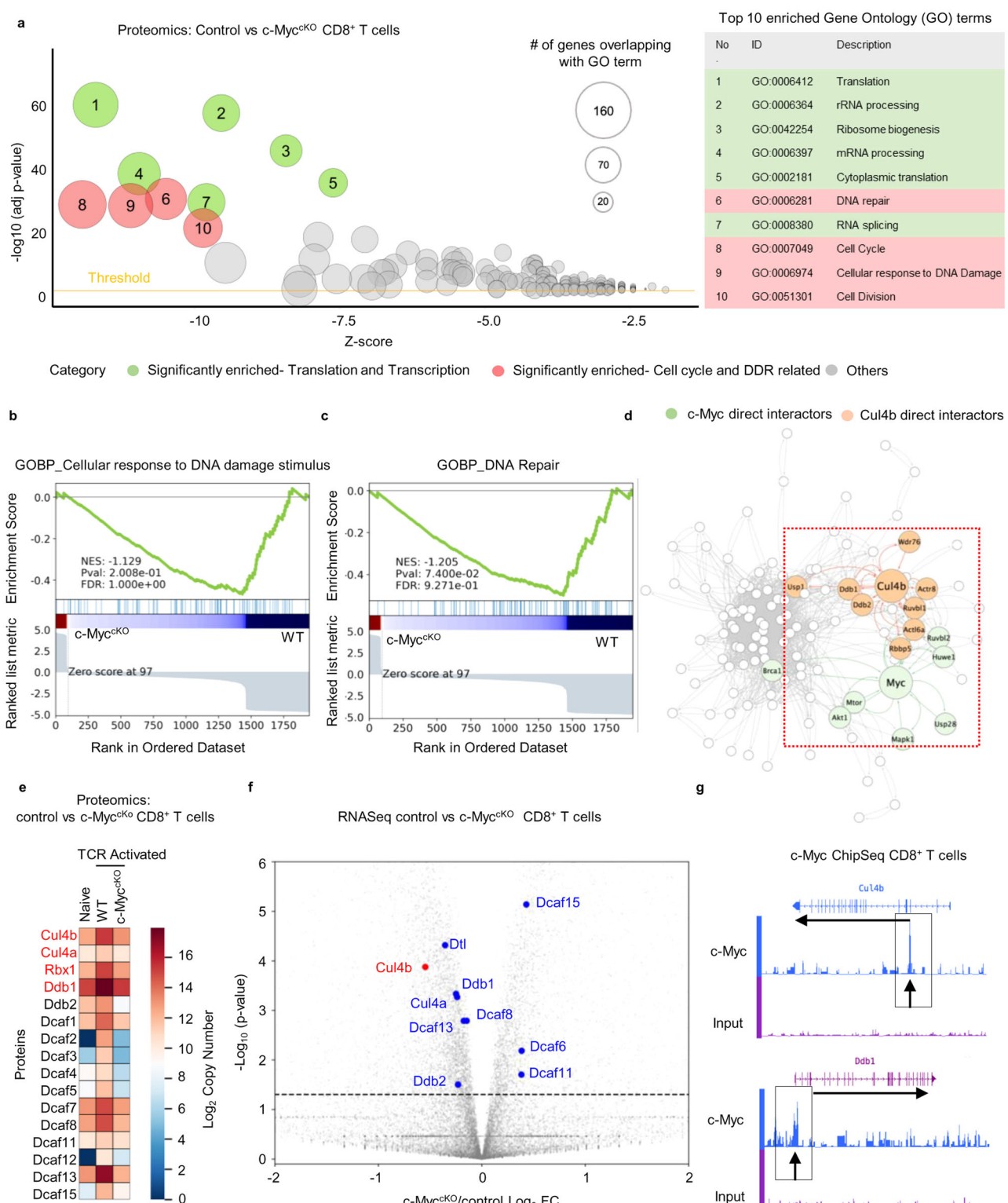

Fig. 1c, d). These observations suggested that c-Myc is a critical factor in the mechanisms that link cell cycle progression and DNA damage response pathways.

We next analyzed the possible protein-protein interaction (PPI) networks among the dysregulated proteins in the control and c-Myc^cKO CD8^+ T cells. The differentially regulated proteins in the c-Myc deficient CD8^+ T cells overlapped with the GOBP categories of cell cycle, DNA repair, and cellular response to DNA damage; these were superimposed with the scored protein interaction data from

STRING and visualized in Cytoscape (v. 3.9.1). The combined interaction scores of 0.750 or above (out of 1.000 maximum) were applied to construct the PPI network. This generated a network of 126 nodes and 1916 interaction edges with PPI enrichment $p$-value $< 1.0 \times 10^{-16}$. Nodes were spatially positioned using Cytoscape's edge-weighted spring embedded layout. Nodes with no interactions with other nodes were filtered out. Interestingly, we found that components of the CRL4B complex and c-Myc were spatially close to each other and were important to sustain network

**Fig. 1 | Cul4b is transcriptionally regulated by c-Myc in CD8+ T cells.**
**a** Quantitative proteomics analysis of c-Myc-deficient (c-Myc^cKO^) and control CD8+ T cells stimulated for 24 h with anti-CD3/CD28 mAbs. The DAVID functional annotation tool was used to perform a gene-annotation enrichment analysis of the set of differentially expressed genes (adjusted p-value < 0.05) between c-Myc^cKO^ and control CD8+ T cells. The bubble plot depicts the enriched functional networks of proteins. The z-score is assigned to the x-axis and the negative logarithm of the adjusted p-value to the y-axis (the higher, the more significant). The size/area of the displayed circles is proportional to the number of genes assigned to the term. Red, significantly enriched GO terms related to cell cycle and the DNA damage response; Green, Translation and Transcription related terms among the top ten terms. GO terms outside top ten are shown in gray. p-values were calculated using two-tailed Fisher's exact tests, corrected for multiple comparisons via Bonferroni correction. **b**, **c** Gene set enrichment analysis (GSEA) showing GO biological processes related to the cellular response to DNA damage and DNA repair in control over c-Myc^cKO^ CD8+ T cells. p-values represent probability of overrepresentation via two-tailed Fisher's exact tests. p-values were corrected for multiple comparisons via the Benjamini-Hochberg procedure. Normalized enrichment score (NES) reflects the degree to which the gene set is overrepresented at the top or bottom of a ranked list of genes, normalized for gene set size and correlations between the gene set and expression dataset. NES was calculated via GSEA. **d** The construction of the Protein-Protein Interaction (PPI) network for proteins associated with biological terms cell cycle, DNA repair, and cellular response to DNA damage were superimposed with the scored protein interaction data from STRING and these were then visualized in Cytoscape (v. 3.9.1). The 1st and 2nd degree interactors of c-Myc are shown, green depicts direct interactors of c-Myc while orange depicts direct interactors of Cul4b. **e** The heatmap showing protein abundance (copy number) of CRL4B components in naïve, TCR activated control (WT) and c-Myc-deleted CD8+ T cells. **f** Naïve CD8+ T cells from control and c-Myc^cKO^ mice were activated with anti-CD3/CD28 plus ICAM1 for 36 h. A volcano plot of gene profiles in c-Myc^cKO^ versus control CD8+ T cells as determined using RNA-seq analysis. Differentially regulated Cul4b components are highlighted. Student's t-tests were performed to identify differentially regulated genes between control and c-Myc^cKO^ samples, p-value < 0.001 were considered significant. **g** c-Myc ChIP-seq tracks at the Cul4b, and adapter protein (Ddb1) gene loci in WT CD8+ T cells (top, gene structures and transcriptional orientations). c-Myc binding peaks, identified by the Model based Analysis for ChIP-seq (MACS) algorithm, are marked by rectangles. Source data are provided as a "Source Data" file.

connectivity (Fig. 1d). We have previously shown that Cul4b is expressed in naïve and activated CD4+ and CD8+ T cells, and its expression and enzymatic activity increased following TCR stimulation[25]. To assess whether this spatial proximity between Cul4b components and c-Myc has any biological relevance, we analyzed the ability of c-Myc to regulate expression of CRL4B components. To do this, we analyzed the absolute protein copies per cell using the 'proteomic ruler' method[32]. We found that in control cells, the relative abundance of key proteins of the CRLB complex, e.g; Cul4b, the adapter protein (DDB1), and the RING protein (RBX1), were increased in TCR activated CD8+ T cells compared to naïve CD8+ T cells (Fig. 1e). Cul4a, an ortholog of Cul4b, also increased after stimulation; however, its levels were significantly lower than Cul4b in activated CD8+ T cells (Supplementary Fig. 1e). Not only were the adapter and RING proteins increased, but most of the substrate receptors for Cul4b were significantly higher in TCR-activated CD8+ T cells compared to naïve CD8+ T cells. Remarkably, the loss of c-Myc in CD8+ T cells prevented upregulation of CRL4B components upon TCR mediated activation (Fig. 1e). Notably, elevated levels of CRL4B components remained high following the differentiation of CD8+ T cells. Both activated (TCR stimulated) and differentiated (Cytotoxic CD8+ T cells) expressed similar levels of CRL4B components and these were significantly higher than levels in naïve CD8+ T cells (Supplementary Fig. 1f). Transcriptome analyses of control and c-Myc^cKO^ CD8+ T cells showed that expression of some of the components of the CRL4 complex were regulated by c-Myc; supporting this, both *Cul4b* and *Ddb1* transcripts were significantly reduced in c-Myc^cKO^ CD8+ T cells (Fig. 1f). Chromatin immunoprecipitation coupled to next-generation sequencing (ChIP-Seq) identified c-Myc binding sites in the loci of *Cul4b, Ddb1 and Rbx1* (Fig. 1g and Supplementary Fig. 1g). These data support that key CRL4B components are direct downstream targets of c-Myc and that c-Myc increases their gene expression following TCR activation. Taken together, these results endorse a model in which c-Myc is upregulating Cul4b and other components of Cul4b complex, providing a direct link between Myc and Cul4b-mediated DDR pathways. To address the functional relevance of c-Myc and its downstream target Cul4b in activated CD8+ T cells, we ablated the expression of Cul4b in mature CD8+ T cells. Deletion of Cul4b was confirmed by immunoblot analyses (Supplementary Fig. 1h). Loss of Cul4b did not impact the cell size of CD8+ T cells after activation (Supplementary Fig. 1i). Notably, we also found that c-Myc induction was largely comparable in control (WT) and Cul4b-deficient (KO) CD8+ T cells rendering it a useful system to dissect the role of c-Myc and its downstream target Cul4b in regulating DDR pathways to regulate antiviral immune responses.

## Loss of Cul4b results in reduced numbers of activated and effector CD8+ T cells

c-Myc has been shown to regulate T cell proliferation and survival upon T cell receptor engagement. To test whether its downstream transcriptional target Cul4b aids CD8+ T cell proliferation, we employed in vitro co-cultures to compare the fitness of control and Cul4b-deficient CD8+ T cells. We co-cultured naïve CD8+ T cells isolated from control (CD45.1+) and Cul4b^cKO^ (CD45.2+) mice in the presence of anti-CD3/CD28 mAbs and assessed cells after 3 or 5 days. At both time points the frequencies of Cul4b-deficient CD8+ T cells were lower than control cells, and differences became more dramatic over time (Fig. 2a, b). Thus, Cul4b was needed to support the dividing CD8+ T cells. Pursuant with this, Cul4b-deficient CD8+ T cells proliferated less than control cells, as evidenced by more loss of Cell Trace Violet in control cells (Fig. 2c, d). Although Cul4b^cKO^ cells were able to proliferate under these conditions, they failed to maintain the rapid proliferation rates of control cells, as quantified by a decreased proliferation index on day 3 (Fig. 2d). Thus, despite a robust induction of c-Myc in Cul4b-deficient CD8+ T cells (Supplementary Fig. 1h), they proliferated at a slower rate, indicating that c-Myc mediated CD8+ T cell proliferation relies on Cul4b. These results support a pathway downstream of TCR signaling and c-Myc that accelerates Cul4b expression to promote CD8+ T cell expansion.

To assess how Cul4b regulates CD8+ T cell function in an intact host, we compared control and Cul4b^cKO^ CD8+ T cell frequencies under homeostatic conditions. To do so, we generated mixed-bone-marrow (BM) chimeras to allow a direct comparison of control and Cul4b^cKO^ CD8+ T cells within the same host. We injected congenically distinct BM cells (T cell-depleted) from control (CD45.1) and Cul4b^cKO^ (CD45.2) mice into sublethally irradiated Rag1^-/-^ mice. Once generated in the host, T cells undergo homeostatic expansion to repopulate the T cell compartment of the host, and these cells were then analyzed 8–10 weeks after BM cell transfer. To normalize for differences in reconstitution, congenically distinct B cell counterparts were used as baseline since B cells would be identical in regards to Cul4b expression. Following reconstitution, we observed that the overall frequencies of Cul4b-deficient CD8+ T cells were significantly lower compared to control CD8+ T cells in the spleen, lymph nodes, and lung of the same host (Fig. 2e and Supplementary Fig. 2a). Cul4b deficiency impacted the frequencies of both activated (CD44+) and naïve (CD44-) CD8+ T cells in lymphoid tissues (Fig. 2f, g and Supplementary Fig. 2b, c). However, in the case of activated CD8+ T cells, the difference between control and Cul4b^cKO^ cells was much greater suggesting that Cul4b deletion was more detrimental to the maintenance of activated T cells than of naïve T cells. To examine the basis of the reduction of

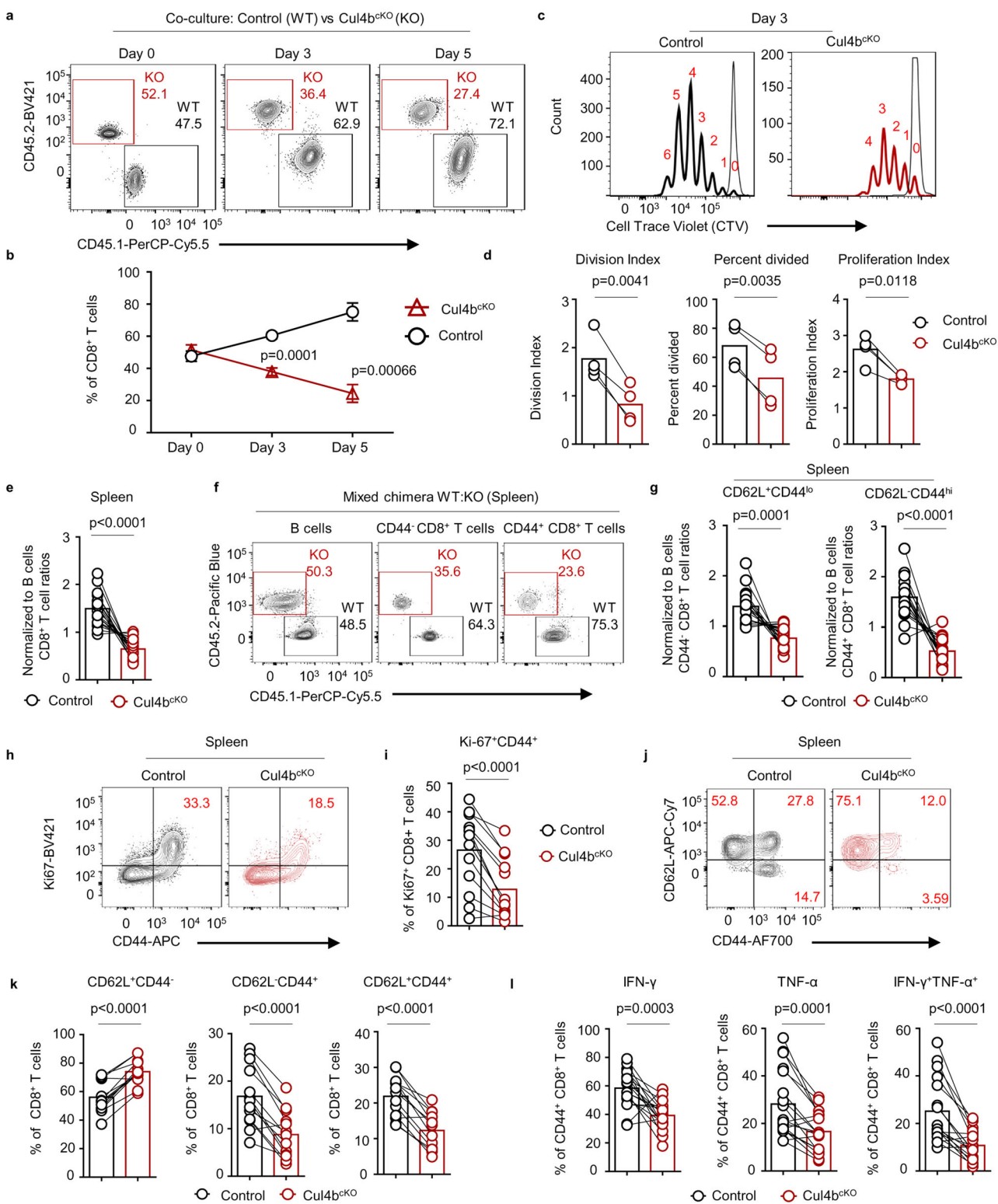

Cul4b$^{cKO}$ T cells, the proportion of cells in cell cycle was assessed using Ki-67, a nucleolar protein not expressed in $G_0$ quiescent cells. Interestingly we observed strikingly lower percentages of Ki-67$^+$CD44$^+$ CD8$^+$ T cells in the spleen and lung of Cul4b$^{cKO}$ mice, an observation consistent with Cul4b regulating T-cell expansion (Fig. 2h, i and Supplementary Fig. 2d, e). For CD8$^+$ T cells, commitment to proliferation is tightly coupled with a commitment to effector- and memory-cell differentiation. CD44 and CD62L (L-selectin) surface markers were used to define the three major subsets of CD8$^+$ T cells in mice. We observed

that compared with co-existing control CD8$^+$ T cells, Cul4b-deficient CD8$^+$ T cells were composed predominantly of naïve (CD62L$^+$CD44$^-$) and were less likely to be effector/memory (CD62L$^-$CD44$^+$) and central memory (CD44$^+$CD62L$^+$) phenotype CD8$^+$ T cells (Fig. 2j, k and Supplementary Fig. 2f, g).

Effector and memory functional properties are imprinted during early stages of CD8$^+$ T cell priming and expansion, we next assessed the functional competence of control and Cul4b$^{cKO}$ effector CD8$^+$ T cells to produce effector cytokines IFN-γ and TNF-α. We enumerated cytokine

**Fig. 2 | Cul4b is required for homeostatic proliferation and function of CD8+ T cells. a** Naïve CD8+ T cells isolated from control mice (CD45.1+) and Cul4b^cKO mice (CD45.2) mice were co-cultured and stimulated in vitro with anti-CD3/CD28 mAbs (5 μg/mL) for 3 or 5 days. The relative proportions of the cells were assessed by flow cytometry. **b** The line graph shows the mean ± S.E.M of percentages of cells of 4–6 independent experiments (n = 6 for day 0 and 3, n = 4 for day 5 for each genotype). p-values were calculated using a paired two-tailed t-test. **c** Naïve CD8+ T cells isolated from control (CD45.1+) and Cul4b^cKO mice (CD45.2+) mice were labeled with Cell Trace Violet (CTV), then co-cultured and stimulated in vitro with anti-CD3/CD28 mAbs (5 μg/mL) for 3 days. Representative flow cytometry histograms of CTV labeled CD8+ T cells is shown. **d** Division index, percent divided, and proliferation index was calculated with FlowJo software. Data of n = 4 for each genotype is shown and each paired sample is connected with a line. p-values were calculated using a paired two-tailed t-test. **e** T cell depleted bone marrow (BM) cells were injected into sublethally irradiated Rag1^-/- recipient mice. Congenically marked BM cells from Cul4b^cKO (CD45.2) and control (CD45.1) mice were mixed as a 1:1 ratio and 2 × 10^6 cells were injected into each recipient mouse (Rag1^-/-). Comparison of the CD8+ T cells in the spleen of irradiated recipient chimeric mice after reconstitution of bone marrow cells from control mice (CD45.1+) and Cul4b^cKO (CD45.2+). Data from three independent experiments are shown (n = 18). Each dot represents a recipient mouse and paired samples are connected with a line. p-values were calculated using a paired two-tailed t-test. **f** Naïve (CD62L^hiCD44^lo) and activated/effector memory like (CD62L^loCD44^hi) CD8+ T cells in the spleen of irradiated recipient chimeric mice after reconstitution. **g** The line graphs show the relative ratios of naïve and activated CD8+ T cells in the spleens of the recipient chimeric mice. The relative ratio of CD8+ T cells was calculated by dividing the percentages of CD8+ T cells with the

percentages of B cells (wild type in all cases) from the respective genotypes. Data from three independent experiments (n = 18 recipient mice). Each paired sample is connected with a line. p-values were calculated using a paired two-tailed t-test. **h, i** CD8+ T cells in the spleen were analyzed for Ki-67 expression. Representative plots show the frequencies of Ki-67 positive CD8+ T cells among control and Cul4b^cKO CD8+ T cells. The line graph shows the relative frequencies of Ki-67+CD8+ T cells in spleen of the recipient mice. Data are presented as line graph and bar represents the mean. from two independent experiments with n = 13 recipient per genotype. Each paired sample is connected with a line. p-values were calculated using a paired two-tailed t-test. **j** The distribution of naïve (CD62L^hiCD44^lo), effector memory like (CD62L^loCD44^hi) and central memory like (CD62L+CD44^hi) CD8+ T cells in the spleen of irradiated recipient chimeric mice after reconstitution. **k** Each bar in the graph shows the mean percentage of naïve (CD62L^hiCD44^lo), effector memory like (CD62L^loCD44^hi) and central memory like (CD62L+CD44^hi) CD8+ T cells in the spleen of recipient mice after reconstitution. Data are presented as line graph from two independent experiments with (n = 13 per genotype). Each paired sample is connected with a line. p-values were calculated using a paired two-tailed t-test. **l** Representative plots of intracellular IFN-γ and TNF-α staining of control and Cul4b^cKO CD8+ T cells within the same mixed bone marrow chimeric mouse. CD44+ CD8+ T cells produce cytokines when incubated in the presence of PMA + Ionomycin. The plots indicate the percent of cytokine producing cells of donor-derived CD8+ T cells. Data are from three independent experiments (n = 18 recipient mice), each bar represents the mean and each paired sample is connected with a line. p-values were calculated using a paired two-tailed t-test. Source data are provided as a "Source Data" file.

producing CD8+ T cells and found that Cul4b-deficient CD8+ T cells showed significantly lower cytokine production than control CD8+ T cells (Supplementary Fig. 2h–j). Polyfunctionality, based on the combined production of IFN-γ and TNF-α, was also significantly reduced in Cul4b-deficient CD8+ T cells (Fig. 2l and Supplementary Fig. 2k). Furthermore, the proliferative defect was not due to the skewing of the TCR repertoire as we found no significant change in the frequency or number of DP thymocytes or TCR Vβ chain usage in Cul4b^cKO mice when compared to control mice (Supplementary Fig. 2l, m). Together, these data demonstrated that Cul4b has a particularly important role in the generation and/or homeostasis of CD44+ CD8+ cytokine producing T cells, cells that have likely been activated via their TCR. Thus, our data thus support that Cul4b has overlapping functions with c-Myc in regulating optimal CD8+ T cell expansion and differentiation. Of note, c-Myc expression is not altered in Cul4b-deficient CD8+ T cells, supporting that c-Myc alone is unable to promote proliferative and cytokine production in Cul4b-deficient CD8+ T cells.

## Cul4b promotes CD8+ T cell mediated antiviral immunity

Cul4b deficiency resulted in a decrease of CD44+CD8+ effector/memory T cells in steady-state, prompting us to explore whether Cul4b plays a role in CD8+ T-cell expansion and effector/memory T cell maintenance following viral infection. We used the acute infection model of Lymphocytic Choriomeningitis Virus (LCMV). In control mice, the spleen size increases following LCMV infection as T and B cells expand to fight infection. In contrast, in Cul4b^cKO mice there was significantly less increase in spleen size when normalized to body weight compared to control mice (Supplementary Fig. 3a). To more directly analyze the expansion and contraction of antigen-specific CD8+ T cells following viral infection, we used tetramers of MHC class I embedded with an immunodominant peptide of LCMV (gp-33). We first assessed cells in the blood and found significantly fewer gp33-specific CD8+ T cells in Cul4b^cKO mice compared with wild-type mice at the peak of the effector response (day 8 post-infection (d8 p.i.) (Fig. 3a, b and Supplementary Fig. 3b). This was also seen at later time points (d16, d24 and d32 p.i.) (Fig. 3a, b). Reductions in virus-specific CD8+ T cells were also seen in the spleen and lymph nodes at the peak of immune response (d8) (Fig. 3c–e) and in spleen at later time points

(d32) (Fig. 3f, g). The defect in the generation of antigenic specific cells was not limited only to CD8+ T cells, we also found that the virus-specific CD4+ T cell response was severely diminished in Cul4b^cKO mice (Supplementary Fig. 3c–f). At the peak of the response, numbers of terminally differentiated short lived effector cells (SLECs) KLRG1+ CD8+ T cells were reduced in Cul4b^cKO compared to control mice (Supplementary Fig. 4a–c). Furthermore, Cul4b^cKO mice showed a profound decrease in effector cells and this was not accompanied by a corresponding increase in the numbers of KLRG1- CD127^hi cells (Supplementary Fig. 4d).

We sought to establish whether Cul4b deficiency might perturb CTL function during viral infection. We first examined the expression of cytolytic granule CD107a protein after in vitro stimulation with gp33 (LCMV gp33-41) or PMA and Ionomycin for 4 h. We found that on peptide stimulation we could not detect any Cul4b^cKO effector CD8+ T cells expressing CD107a compared to control effector CD8+ T cells (Supplementary Fig. 4e, f). This may be because the antigenic specific Cul4b^cKO CD8+ T cell numbers were below the level of detection. Thus, we examined CD107a expression in response to PMA/ionomycin stimulation and found that CD107a+ cells among Cul4b-deficient CD8+ T cells were greatly reduced (Supplementary Fig. 4g, h). We next examined the ability of control and Cul4b^cKO CD8+ T cells to produce IFN-γ and TNF-α after 4 h in vitro stimulation with peptide and PMA/Ionomycin (Supplementary Fig. 4i–l). Cul4b-deficient CD8+ T cells had greatly reduced production of IFN-γ, and TNF-α, (Supplementary Fig. 4j, l). Notably, the percentages of polyfunctional, dual cytokine-producing (IFN-γ, TNF-α) Cul4b^cKO CD8+ T cells were also profoundly decreased. These defects in T cell responses resulted in a severe defect in viral control as mice lacking Cul4b in T cells continued to show high titers of virus at a time point when most control mice had cleared infection (Fig. 3h).

Similar observations were seen following infection of mice with a murine tropic virus. Interferons (IFNs), CD4+ and CD8+ T cells are important in the recovery from infection with ectromelia, a mousepox virus[33,34]. Cul4b^cKO mice infected with ectromelia displayed increased inflammation (footpad swelling) and foot loss (Fig. 3i, j). These data indicated that the requirement for Cul4b in CD8+ T cell expansion was not limited to LCMV infection but could be a general feature of viral infections. These data also suggested that in the absence of Cul4b,

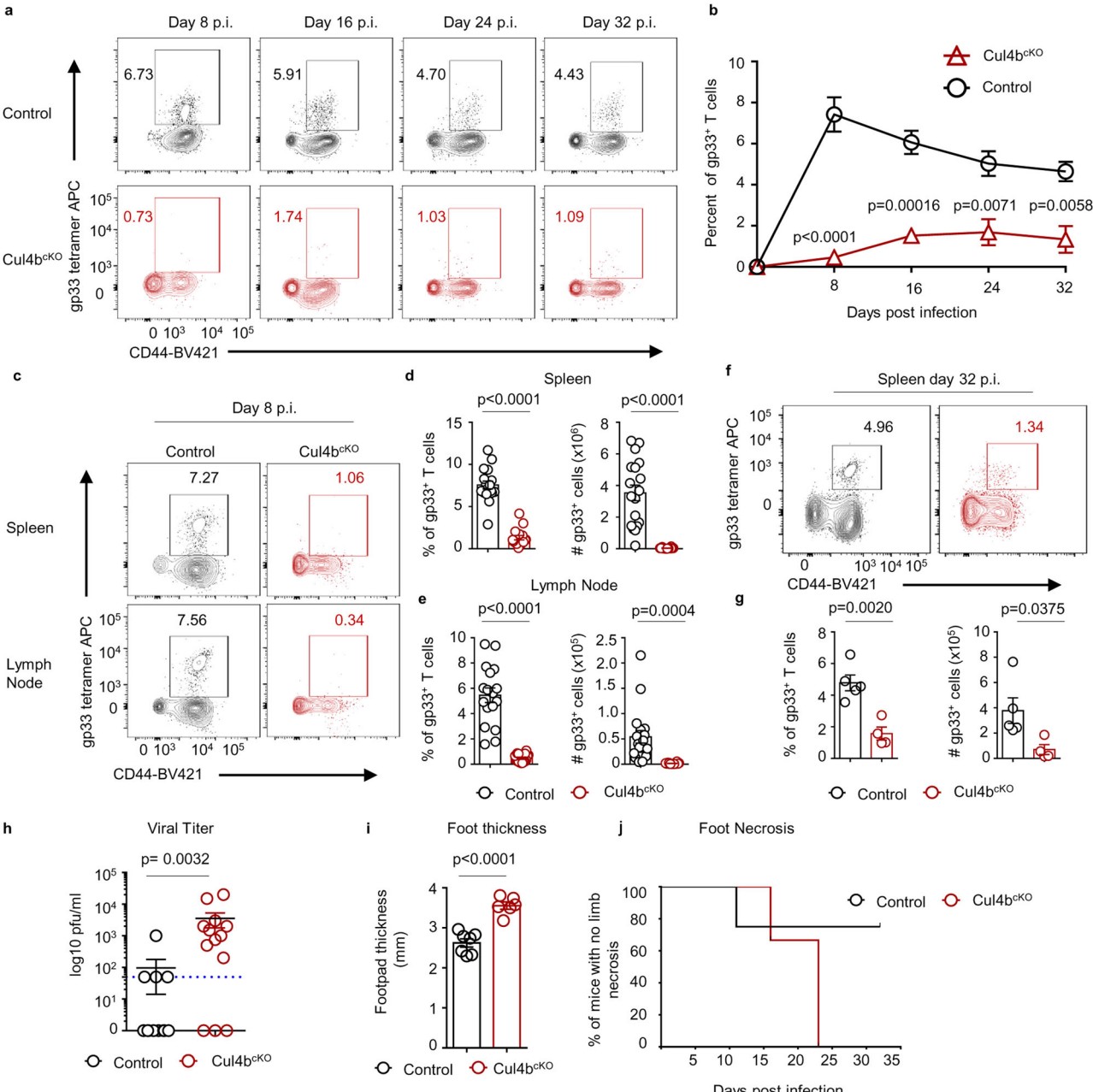

**Fig. 3 | Cul4b is required for the expansion of Ag-specific CD8⁺ T cells following viral infection.** Cul4b^{cKO} and control mice (C57BL/6 littermates) were intraperitoneally infected with LCMV Armstrong strain (2 × 10^5 pfu) and analyzed at the indicated time points. **a** Flow cytometry plots gated on CD8⁺ T cells showing percentages of gp33-specific cells at d8, d16, d24 and d32 p.i. in the blood. **b** The percentages of gp33-specific CD8⁺ T cells in the blood is shown as Mean ± S.E.M. Data are representative of two independent experiments with $n = 4$ for control and $n = 5$ for Cul4b^{cKO} genotypes. $p$-values were calculated using an unpaired two-tailed $t$-test. **c–e** The percentages and numbers of gp33-specific CD8⁺ T cells at d8 p.i. in the spleen and lymph nodes are shown as mean ± S.E.M. Data are cumulative of three independent experiments with $n = 17$–$18$ for control and $n = 16$ for Cul4b^{cKO} group. $p$-values were calculated using an unpaired two-tailed $t$-test. **f, g** The percentages and numbers of gp33-specific CD8⁺ T cells at d32 p.i. in the spleen. Data are representative of two independent experiments and is shown as a mean ± S.E.M for $n = 5$ for control and $n = 4$ for Cul4b^{cKO} group. $p$-values were calculated using an

unpaired two-tailed $t$-test. **h** Viral titers in the blood at d8 p.i. shown as a median ± S.E.M. Data are from two independent experiments ($n = 12$ for control and $n = 13$ for Cul4b^{cKO} genotypes. $p$-values were calculated using a two-tailed Mann–Whitney $U$-test. Each symbol represents an individual mouse. **i, j** Cul4b^{cKO} and control mice (C57BL/6 littermates) were infected with ectromelia virus (1 × 10^3 pfu) in their footpad and analyzed at the indicated time points for signs of footpad swelling and limb necrosis. **i** Footpad swelling was measured by calculating the thickness of the infected foot pad; the uninfected foot pad was used as a reference. The data are from two independent experiments ($n = 7$ for control and $n = 6$ for Cul4b^{cKO}). Data are shown as mean ± S.E.M and $p$-values were calculated using an unpaired two-tailed $t$-test. **j** Infected limbs were recorded daily and scored by evaluating the hind limb swelling or presence of tissue necrosis. The data are shown as median survival and are representative of two independent experiments with $n = 4$ for control and $n = 3$ for Cul4b^{cKO}. Source data are provided as a "Source Data" file.

CD8[+] T cells showed a severely defective clonal proliferation, ability to attain effector functions and were ineffective at driving antiviral immunity.

## Cul4b functions in a cell intrinsic manner during the early expansion of CD8[+] T cells

It is unclear whether the effector CD8[+] T cell defects described above were due to a role for Cul4b in CD8[+] T cells or secondary effects of failed viral control. To distinguish between these two possibilities, congenically distinct wild-type (CD45.1/2) and Cul4b[cKO] (CD45.2) P14 CD8[+] T cells (specific for the LCMV gp33 epitope presented on H-2D[b]) were mixed at a 1:1 ratio and adoptively transferred into wild-type (CD45.1) recipients. One day later, the mice were infected with LCMV Arm. We found that by d8 p.i. the number of Cul4b[cKO]P14 CD8[+] T cells was profoundly decreased compared to control P14 cells (Fig. 4a). Cul4b[cKO] CD8[+] T cells did not expand as effectively as the control cells, resulting in a 40-fold decrease in the relative proportion of Cul4b[cKO] CD8[+] T cells. The decreased frequency of Cul4b[cKO] P14 cells was observed in multiple lymphoid organs (Fig. 4b and Supplementary Fig. 5a, b). Thus, despite having the same access to antigen presenting cells, CD4 help, and cytokines as control CD8[+] T cells, Cul4b[cKO] CD8[+] T cells failed to expand following LCMV infection.

CD8[+] T cells enter into cell cycle ~2 days following infection. To determine how early is this defect in expansion became evident in Cul4b[cKO] P14 T cells, we analyzed the relative distribution of control and Cul4b[cKO] P14 T cells in recipient mice by d2 and d5 post infection. By d2 post infection, equal percentages of control and Cul4b[cKO] P14 T cells were recovered from the recipient mice (Fig. 4c, d). However, by d5 post infection there were significantly fewer Cul4b[cKO] compared to control (WT) P14 T cells (Fig. 4e, f). These data were consistent with a loss of Cul4b-deficient CD8[+] T cells just as cells were starting to proliferate. Furthermore, when we directly transferred purified control P14 or Cul4b[cKO] P14 CD8[+] T cells into congenic WT recipients (Supplementary Fig. 5b) we saw similar results, with few Cul4b-deficient cells evident during either the peak of the CD8[+] T cell response (day 8) or at later time points (day 16 and 32) (Fig. 4g, h and Supplementary Fig. 5c). This lack of expansion of Cul4b[cKO] P14 CD8[+] T cells was also observed in the spleen (Supplementary Fig. 5d, e). These data support that the expansion of CD8[+] T cells in the first few days after they encounter APCs loaded with viral peptide requires Cul4b.

## Cul4b is required for the formation of memory CD8[+] T cells

It was recently shown that c-Myc segregates asymmetrically in the two daughter cells generated during this first division to determine the effector vs. memory fate. Specifically, c-Myc preferentially moves to the proximal daughter to promote the effector fate. Our data supported that Cul4b is a target of c-Myc and that Cul4b deficient CD8[+] T cells were unable to expand to generate sufficient numbers of effector cells. Thus, we next sought to test whether Cul4b is also required for the development of memory CD8[+] T cells.

We first assessed whether Cul4b expression was differentially regulated between different subsets of effector and memory CD8[+] T cells during LCMV infection. We found that c-Myc and various components of Cul4b complex were expressed in multiple subsets of CD8[+] T cells: Long Lived effector cells (LLE), Effector Memory (T_EM) and Central memory (T_CM) (Fig. 5a). Subsets were confirmed by analyzing expression of distinguishing characteristic markers CX3CR1 and CXCR3 (Supplementary Fig. 6a). We found that Cul4b levels were similar between effector and memory CD8[+] T cells, leading us to posit that it might also be required for the formation of memory cells. Thus, we infected control and Cul4b[cKO] mice and analyzed virus-specific memory cells. During acute infection, KLRG1[hi]IL-7R[lo]CD8[+] T cells have been identified as short-lived effector cells (SLEC), whereas KLRG1[lo]IL-7R[hi] cells give rise to long-lived memory T cells[35]. Analysis of LCMV infected mice revealed that ~50% of control gp33[+]CD8[+] T cells in the

spleen at d32 p.i. expressed IL-7R (CD127) (Fig. 5b, c), while Cul4b[cKO] gp33[+]CD8[+] T cells lacked expression of both KLRG1 and IL-7R (Fig. 5b, c). Approximately 15% of the control antigen-specific CD8[+] T cells were CD62L[+] T central Memory (T_CM) cells, while Cul4b[cKO] gp33[+]CD8[+] T cells showed a significant reduction in percentages and numbers of these cells (Fig. 5d, e). In contrast, Cul4b[cKO] gp33[+]CD8[+] T cells expressed PD-1, a marker of T cell exhaustion (Fig. 5f). Further evidence of exhaustion included decreased lytic potential and reduced cytokine production (Fig. 5g and Supplementary Fig. 6b–g). These data are consistent with our observation that Cul4b[cKO] mice fail to clear virus, making these mice poor models for studies of Cul4b in T cell memory.

To test the role of Cul4b in the formation of memory CD8[+] T cells, congenically distinct Cul4b[cKO] and control P14 CD8[+] T cells were mixed 1:1 and transferred into naïve wild-type recipients (Supplementary Fig. 6h). One day later, mice were infected with LCMV. d32 p.i. P14 CD8[+] T cells were evaluated for their expression of CD62L, KLRG1 PD-1 and CD127. The numbers and percentages of Cul4b[cKO] P14 T cells recovered were profoundly decreased compared to controls (Fig. 5h, i and Supplementary Fig. 6i). Furthermore, when transferred together with WT P14 cells, Cul4b-deficient did not show elevated levels of PD-1 (Supplementary Fig. 6j). This latter finding supported that high PD-1 expression observed in Fig. 5f was due to reduced viral clearance. Upon phenotypic characterization, we found that Cul4b[cKO] P14 CD8[+] T cells were much less likely to be CD62L[hi] compared to control P14 cells (Fig. 5j, k). This suggested that either Cul4b is required for memory development or its expression in CD8[+] T cells provides a competitive advantage to memory cells. To distinguish between these two possibilities, we adoptively transferred control or Cul4b[cKO] P14 cells into separate recipient mice and again analyzed cells at d32 p.i. (Fig. 5l, m). We again found a significant decrease in Cul4b[cKO] P14 CD8[+] memory T cells (Fig. 5m). Thus, Cul4b plays an important role in the expansion of both memory and effector CD8[+] T cells to promote antiviral immunity.

## Cul4b regulates the DNA damage response and cell cycle progression in CD8[+] T cells

Our finding that Cul4b was required for CD8[+] T cell expansion and antiviral immunity prompted us to investigate underlying molecular mechanisms. To explore how Cul4b impacts proteome remodeling during immune activation, we performed quantitative label-free high-resolution mass spectrometry on control and Cul4b[cKO] CD8[+] T cells stimulated with anti-CD3/CD28 mAbs for 40 h. This time point allowed maximal expression of Cul4b with little or no difference in survival between control and Cul4b[cKO] T cells. Greater than 6400 proteins were identified and among them 450 proteins were significantly dysregulated (Supplementary Fig. 7a). The dysregulated proteins were functionally annotated using the ClueGO bioinformatics tool to identify biological processes regulated by Cul4b in CD8[+] T cells following activation. For proteins that were upregulated in Cul4b-deficient CD8[+] T cells, ClueGO revealed specific terms like 'Cell Cycle process, DNA-dependent DNA replication, Chromosome organization and others liked to DNA metabolism (Supplementary Fig. 7b). The GOplot algorithm was used to visualize the biological processes and these are depicted in the GOCircle plot. We found protein clusters associated with DNA repair, cell cycle, DNA damage response were among the top ten most enriched clusters (Supplementary Fig. 7c). These observations were consistent with c-Myc deletion in CD8[+] T cells and suggested that Cul4b functions downstream of c-Myc to control the cell cycle progression and DNA damage response pathways.

We hypothesized that Cul4b might aid the recruitment of proteins to chromatin during cell cycle progression and in response to DNA damage. To investigate this, we first analyzed the ability of Cul4b to bind chromatin in proliferating CD8[+] T cells. We analyzed the relative abundance of Cul4b in soluble and chromatin bound fractions. CD8[+] T cells were stimulated for 40 h and soluble fraction (SF) and

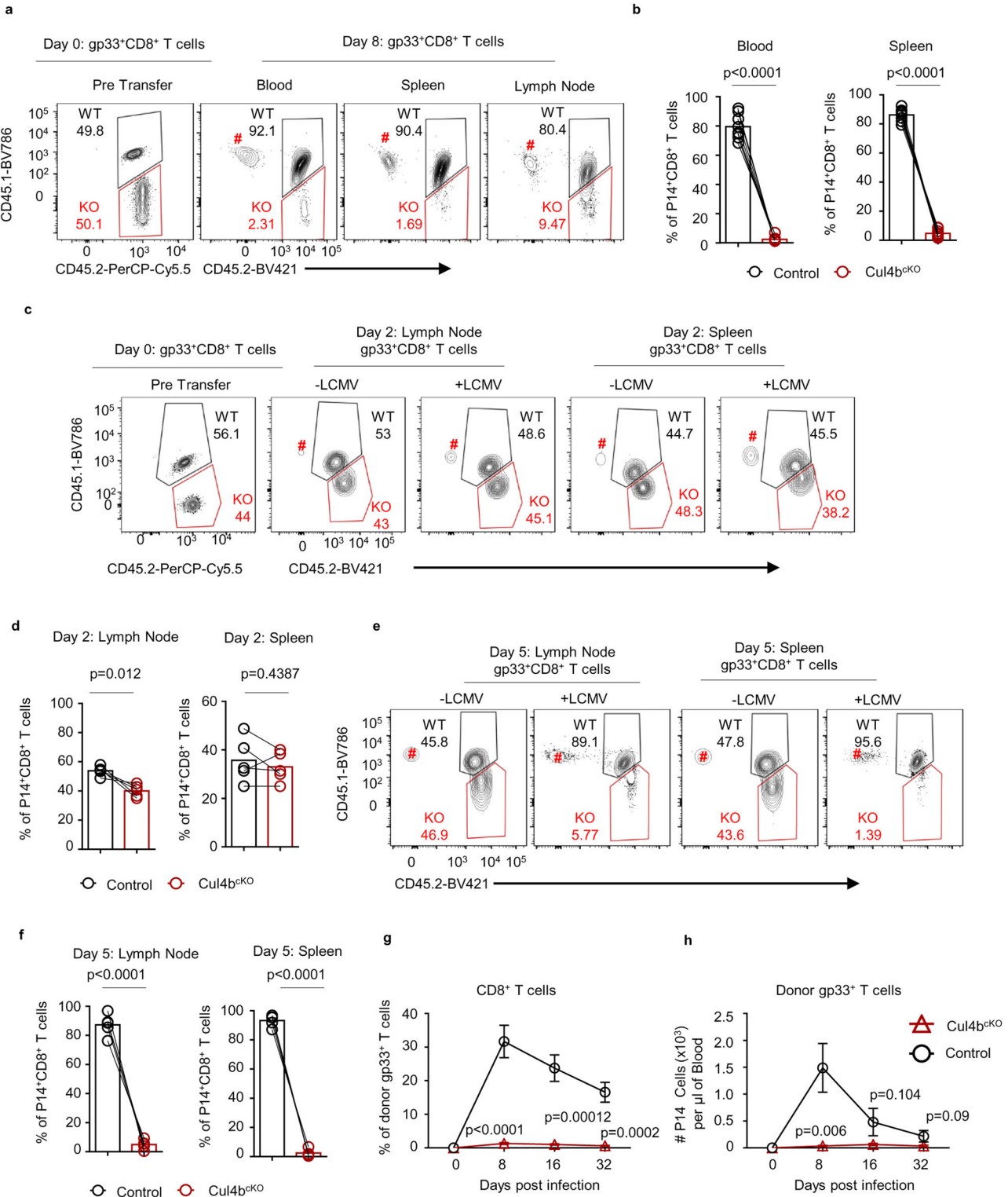

chromatin bound fraction (CBF) were isolated. Bands corresponding to both neddylated and un-neddylated forms of Cul4b were found in the soluble fraction (Fig. 6a). In contrast, the Cul4b bound to chromatin was almost exclusively in the neddylated (active) form (Fig. 6a). Supporting that the slower migrating band was indeed neddylated Cul4b, addition of MLN4924, a Nedd8 activating enzyme (NAE) inhibitor, triggered a rapid loss of this band in both the soluble fraction and chromatin bound fraction. To investigate whether Cul4b impacts chromatin-association of proteins, we used mass spectrometry to compare the abundance of proteins associated with chromatin in

control and Cul4b[cKO] CD8[+] T cells following DNA damage (Fig. 6b). Specifically, Camptothecin was used to induce DNA damage. This would allow us to normalize for differences in DNA damage between the control and Cul4b[cKO] CD8[+] T cells. More than 6000 proteins were identified, and among these 1100 proteins were significantly different between control and Cul4b[cKO] CD8[+] T cells (Supplementary Fig. 8a). Cul4b and c-Myc copy numbers in TCR activated (40 h) CD8[+] T cells were calculated using the proteomic ruler method[32] (Supplementary Fig. 8b). We used the Molecular signature Database (MsigDB) to identify hallmark gene sets associated with the proteins upregulated in

**Fig. 4 | Cul4b regulates CD8⁺ T cell expansion early after antigen encounter in a cell-intrinsic manner.** Congenically distinct Cul4b$^{cKO}$ (CD45.2) and control (CD45.1/CD45.2) P14 cells (total $5 \times 10^4$ - $1 \times 10^5$ cells) were mixed at a 1:1 ratio and adoptively transferred into naïve recipients (CD45.1). Recipient mice were infected intraperitoneally with LCMV ($2 \times 10^5$ pfu) and analyzed at the indicated time-points. **a** Flow cytometry plots gated on P14 cells showing percentages of wild-type (upper, CD45.1⁺CD45.2⁺) and Cul4b$^{cKO}$ (lower, CD45.2⁺) at the time of adoptive transfer and d8 p.i. in the blood, spleen and lymph nodes. **b** Percentages of Cul4b$^{cKO}$ and control P14 cells in the blood and spleen at day 8 p.i. Data from $n = 8$ mice for each genotype is shown and each paired sample is connected with a line. *p*-values were calculated using a paired two-tailed *t*-test. **c** Flow cytometry plots gated on P14 cells showing percentages of wild-type and Cul4b$^{cKO}$ at the time of adoptive transfer and d2 p.i. in the spleen and lymph nodes. **d** Percentages of Cul4b$^{cKO}$ and control P14 cells in the

lymph node and spleen at day 2 p.i. Data from $n = 5$ mice for each genotype is shown and each paired sample is connected with a line. *p*-values were calculated using a paired two-tailed *t*-test. **e, f** Flow cytometry plots gated on P14 cells showing percentages of wild-type and Cul4b$^{cKO}$ at the time of adoptive transfer and d5 p.i. in the lymph node and spleen. #Represents the gp33-specific CD8 + T cells from the recipient mice. Data from $n = 5$ mice for each genotype is shown and each paired sample is connected with a line. *p*-values were calculated using a paired two-tailed *t*-test. **g, h** Separate transfers of Cul4b$^{cKO}$ and control P14 cells ($1 \times 10^5$ cells) showed a defective Cul4b$^{cKO}$ CD8 T cell response after LCMV infection. The percentages and numbers of P14 cells per µL of blood are shown. Data are representative of two independent experiments ($n = 5$ recipients were used for each genotype). Data is shown as mean ± S.E.M and *p*-values were calculated using an unpaired two-tailed *t*-test. Source data are provided as a "Source Data" file.

Cul4b-deficient cells. This revealed an enrichment of hallmarks associated with the G2-M checkpoint and Myc targets among others (Fig. 6c). c-Myc deletion in CD8⁺ T cells showed an enrichment of similar hallmark gene sets compared to Cul4b deletion (Supplementary Fig. 8c). GSEA also showed hallmarks gene sets linked to G2-M Checkpoint and Myc targets as well as the p53 pathway were enriched in Cul4b deficient CD8⁺ T cells compared to control cells (Fig. 6d, e and Supplementary Fig. 8d). The strong signature of G2-M checkpoint and Myc targets supported that c-Myc mediates cell cycle progression and DNA damage response pathways through upregulation of Cul4b. We then functionally annotated the upregulated proteins using the ClueGO Reactome bioinformatics tool. Pathway analysis was performed by calculating the gene set enrichment analysis score for each pathway. The pathway annotations revealed that the top two Reactome hits belong to aberrant regulation of mitotic G1/S transition and activation of the pre-replicative complex (Fig. 6f). Using GSEA, we found that proteins associated with these two pathways were enriched in Cul4b-deficient CD8⁺ T cells (Fig. 6f and Supplementary Fig. 8e). Overall, these data indicated that Cul4b and c-Myc are directly involved in the regulation of cell cycle checkpoints in proliferating CD8⁺ T cells.

We next analyzed the potential protein-protein interaction (PPI) networks among the upregulated proteins in the Cul4b$^{cKO}$ CD8⁺ T cells over control cells. The differentially upregulated proteins in the Cul4b$^{cKO}$ CD8⁺ T cells along with Cul4b and c-Myc were superimposed with the scored protein interaction data from STRING and visualized in Cytoscape (v. 3.9.1) as described previously. Cul4b and c-Myc were added to reveal potential interactions with the deregulated proteins. We identified unique 1$^{st}$ and 2$^{nd}$ degree interactors of Cul4b, unique 1$^{st}$ and 2$^{nd}$ degree interactors of c-Myc and mutual 1$^{st}$ and 2$^{nd}$ degree interactors of both Cul4b and c-Myc (Fig. 6h). Interestingly, the mutual interactors included replication licensing factors, G1/S transition or G2/M transition and DNA damage regulators. Among the common interactors, Cyclin E2 (Ccne2) stood out as it was one of the most differentially expressed proteins between Cul4b$^{cKO}$ and control CD8⁺ T cells (Fig. 6i). This was particularly interesting given that oncogenic activation of Cyclin E/CDK2 impairs DNA replication, causing replication stress and DNA damage. In addition, p21 (Cdkn1a), a negative regulator of Cyclin E/CDK2, was also among the 1$^{st}$ and 2$^{nd}$ degree interactors of both Cul4b and c-Myc. As an additional validation, we overlapped these results with proteins identified in the whole cell proteome and chromatin enriched proteomes of CD8⁺ T cells and found significant overlap between the two datasets (Fig. 6j). Comparison of the log$_2$ fold changes of the common proteins showed high concordance, with pair-wise Pearson's correlation coefficients of 0.8 (Fig. 6j). Overall, these analyses support that Cul4b is required for the regulation of cell cycle checkpoints in proliferating CD8⁺ T cells to avoid replication stress. By raising the levels of Cul4b, c-Myc grants protection against replicative stress in rapidly dividing CD8⁺ T cells, ensuring rapid repair of DNA damage and linking repair pathways to cellular proliferation.

## Cul4b circumvents replication stress in CD8⁺ T cells by controlling levels of p21 and Cyclin E2

The tumor suppressor and transcriptional regulator p53 is stabilized in response to DNA damage and regulates the expression of numerous target genes including p21. However, the role of p21 in cell cycle control in CD8⁺ T cells is less clear. To assess whether Cul4b regulates p21 in the presence and absence of DNA damage, we treated activated Cul4b$^{cKO}$ and control CD8⁺ T cells with Camptothecin for 2 h. Specifically, control (WT) and Cul4b$^{cKO}$ (KO) cells were activated for 40 h and Camptothecin was added to the cultures for the last 2 h. To examine the abundance and distribution of p53 and p21, we fractionated CD8⁺ T cells into a chromatin bound, and soluble protein fraction as described previously. This analysis confirmed that the majority of p53 was associated with chromatin and its levels increased following DNA damage. However, the levels of p53 were more or less similar between control and Cul4b$^{cKO}$ CD8⁺ T cells, both in the presence and absence of DNA damage (Fig. 7a). In contrast, levels of p21 accumulated significantly in Cul4b deficient cells, in both the presence and absence of DNA damage (Fig. 7a). A significant proportion of p21 was present in the soluble fraction. Elevated expression of p21 in a p53 neutral environment has been shown to cause genomic instability through deregulation of the replication licensing machinery[36]. Supporting this, we did pair-wise Pearson's correlation to identify overlap between p21 overexpressing[36] and Cul4b$^{cKO}$ proteomic datasets. Comparison of the log$_2$ fold changes of the common proteins shows high concordance, with pair-wise Pearson's correlation coefficients of 0.8 (Fig. 7b). This analysis revealed that p21 accumulation in Cul4b-deficient CD8⁺ T cells could result in deregulation of the replication licensing machinery (Cdt1, Cdc6, and Orc1). Activation of p21 triggers replication stress in a Cdt1/Cdc6-dependent manner, leading to DNA damage and DNA damage response (DDR) activation[36,37]. We assessed DNA damage in activated CD8⁺ T cells after viral infection. Cul4b$^{cKO}$ CD8⁺ T cells had higher H2AX phosphorylation (γ-H2AX) and phosphorylated ATM (pATM) than control CD8⁺ T cells (Fig. 7c and Supplementary Fig. 9a). We further analyzed whether activated Cul4b$^{cKO}$ CD8⁺ T cells were more sensitive to Camptothecin, a DNA damage inducing agent. We found that activated Cul4b-deficient CD8⁺ T cells, that were treated with Camptothecin, displayed significantly higher amounts of DNA damage when assessed using a comet assay (Fig. 7d, e). Together, these data supported that Cul4b-deficiency resulted in increased p21 stability, resulting in DNA damage and activation of the DNA damage response.

This led us to examine what might be causing Cul4b deficient CD8⁺ T cells to enter into S-phase when p21 levels were measurably elevated. Oncogenes linked to the induction of replication stress are the transcription factor c-Myc and Cyclin E[38], which acts collaboratively with cyclin-dependent kinase-2 (CDK2) to promote S-phase entry. Given that c-Myc levels were similar between control and Cul4b$^{cKO}$ CD8⁺ T cells, we turned our efforts to Cyclin E. Our proteomics data showed that Cyclin E2, but not Cyclin E1, was significantly higher in Cul4b$^{cKO}$ CD8⁺ T cells. Thus, we assessed Cyclin E2 levels in

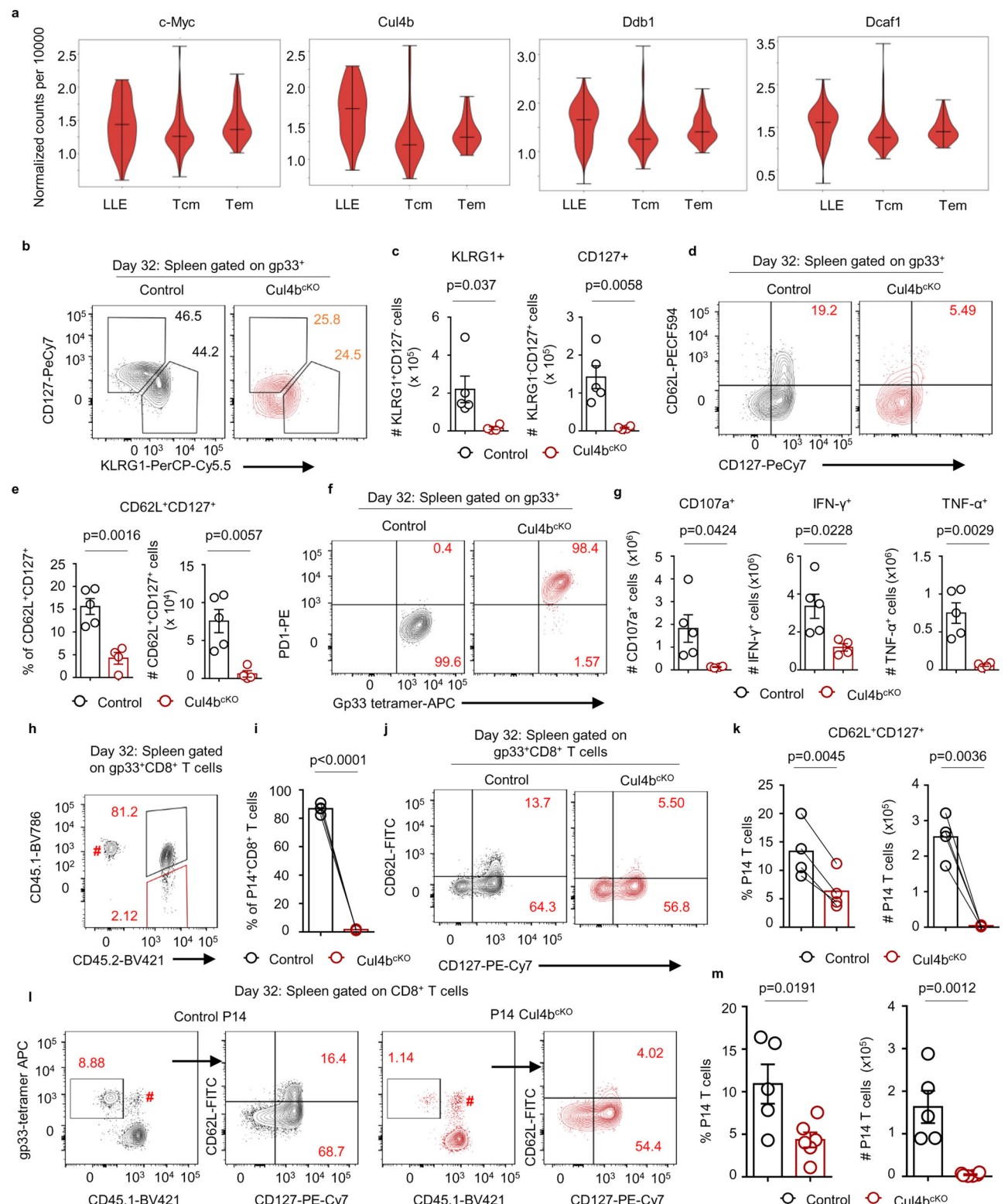

control and Cul4b^cKO CD8+ T cells. Western blotting analysis showed increased levels of Cyclin E2 in Cul4b-deficient CD8+ T cells both in the absence and presence of DNA damage (Fig. 7f, g). In contrast, Cyclin E1 levels were similar in control and Cul4b-deficient CD8+ T cells (Supplementary Fig. 9b). To investigate whether Cyclin E2 was regulated in a proteasome dependent manner, we examined Cyclin E2 protein levels in the presence or absence of the proteasomal inhibitor Bortezomib. We activated CD8+ T cells for 40 h and treated the cells with bortezomib or vehicle for the last 4 h. Bortezomib treatment

significantly decreased the turnover of Cyclin E2 levels in control cells (Fig. 7h). Further, we found no significant difference in transcript levels of either Cyclin E1 (*Ccne1*) or Cylin E2 (*Ccne2*) in Cul4b-deficient CD8+ T cells compared to control cells (Supplementary Fig. 9c). This led us to speculate that Cul4b mediated proteasomal degradation promotes Cyclin E2 protein turnover. Of note, *p53* and *c-Myc* transcript levels were similar between control and Cul4b^cKO CD8+ T cells, while *p21* levels were significantly higher in Cul4b-deficient CD8+ T cells. Thus, Cyclin E2 accumulation in Cul4b-deficient CD8+ T is likely to be the

**Fig. 5 | Loss of Cul4b results in a diminished expansion of central memory CD8+ T cells.** Bulk RNA-seq data from subsets of long-lived effector cells (LLE;CD62L$^{lo}$IL-7Rα$^{lo}$), T effector memory (TEM; CD62L$^{lo}$IL-7Rα$^{hi}$), and T central memory (TCM;CD62L$^{hi}$IL-7Rα$^{hi}$) cells[68]. The subsets were sorted at D35 post LCMV infection. **a** The gene-expression pattern of c-Myc, Cul4b, Ddb1 and Rbx1 in LLE, TEM, and TCM. The violin plot shows the density of the data, range, median, and interquartile range (IQR) of the data. **b** Cul4b$^{cKO}$ and control mice (C57BL/6 background) were infected intraperitoneally with LCMV (2 × 10$^5$ pfu) and analyzed at day 32 post infection for the expression of CD127, KLRG1, and CD62L. **b, c** gp33+ CD8+ T cells were analyzed for CD127 and KLRG1 and the numbers of CD127$^{lo}$KLRG1$^+$ and CD127$^+$KLRG1$^{lo}$ are shown. Data from n = 5 for control and n = 4 for Cul4b$^{cKO}$ genotypes is shown as mean ± S.E.M. p-values were calculated using an unpaired two-tailed t-test. **d, e** Flow cytometry plots show the CD8+ central memory T cell subset as CD62L$^+$CD127$^+$ in Cul4b$^{cKO}$ and control mice. **e** The percentages and numbers of antigen specific CD62L$^+$CD127$^+$ cells in Cul4b$^{cKO}$ and control mice at day 32 post infection. Data from n = 5 for control and n = 4 for Cul4b$^{cKO}$ genotypes is shown as mean ± S.E.M. p-values were calculated using an unpaired two-tailed t-test. **f** Expression of PD-1 on LCMV-specific CD8+ T cells from the spleen of Cul4b$^{cKO}$ and control mice at day 32 post infection. **g** The percentages of CD8+ T cells expressing CD107a, IFN-γ and TNF-α from LCMV infected Cul4b$^{cKO}$ and control mice following stimulation with the PMA/Ionomycin. Representative data from two independent

experiments are shown with mean ± S.E.M, n = 5 for control and n = 4 for Cul4b$^{cKO}$. p-values were calculated using an unpaired two-tailed t-test. **h** Congenically distinct Cul4b$^{cKO}$ and control P14 cells (total 5 × 10$^4$ - 1 × 10$^5$ cells) were mixed at a 1:1 ratio and adoptively transferred into naïve recipients. The recipient mice were infected intraperitoneally with LCMV (2 × 10$^5$ pfu) and analyzed at day 32 post infection. Flow cytometry plots gated on P14 cells showing percentage of wild-type (upper, CD45.1$^+$CD45.2$^+$) and Cul4b$^{cKO}$ (lower, CD45.2$^+$) at adoptive transfer and d32 p.i. in the spleen. #Denotes the P14 cells from the recipient mice. **i** Percentages of Cul4b$^{cKO}$ and control P14 cells in the spleen at day 32 p.i. Data from n = 4 for each genotype is shown and each paired sample is connected with a line. p-values were calculated using a paired two-tailed t-test. **j, k** Flow cytometry plots show the CD62L$^+$CD127$^+$ P14 cells in Cul4b$^{cKO}$ and control mice at day 32 post infection. **k** The percentages and numbers of CD62L$^+$CD127$^+$ P14 cells. Data from n = 4 for each genotype is shown and each paired sample is connected with a line. p-values were calculated using a paired two-tailed t-test. **l, m** Cul4b$^{cKO}$ and control P14 cells (1 × 10$^5$ cells) were transferred into separate recipients and then evaluated for CD62L$^+$CD127$^+$ P14 cells. The percentages and numbers of P14 cells in the spleen is shown. Data from n = 5 for control and n = 6 for Cul4b$^{cKO}$ is shown as mean ± S.E.M. p-values were calculated using an unpaired two-tailed t-test. #Indicates the gp33-specific CD8+ T cells from the recipient mice. Source data are provided as a "Source Data" file.

source of replication stress induced DNA damage apparently fueled by deregulation of replication licensing machinery by the accumulation of p21. Cell sensing of DNA damage initiates cell cycle arrest at the G2-M phase to allow repair of DNA lesions. We found that Cul4b deficient CD8+ T cells were much more likely to be arrested in G2-M phase and were cycling slower than their control counterparts (Fig. 7i and Supplementary Fig. 9d). Cul4b deficient CD8+ T cells also showed signs of re-replication (Fig. 7j and Supplementary Fig. 9e) and increased apoptosis (Fig. 7k and Supplementary Fig. 9f). If the primary role of Cul4b in CD8+ T cells is to regulate DNA damage response to allow for rapid division, then cells that do not divide may be less dependent on Cul4b. To test this, we used Rapamycin or Nocodazole to block anti-CD3/CD28 mAb activated CD8+ T cells in G1 phase or G2-M phase, respectively. We found that both Rapamycin and Nocodazole abolished proliferation and rescued Cul4b-deficient CD8+T cells (Fig. 7l, m). In addition, when cell proliferation was blocked using rapamycin, there was no difference in cell death between control and Cul4b-deficient CD8+ T cells (Supplementary Fig. 9g, h). This indicated that Cul4b functions to prevent replication stress to allow proliferation of CD8+ T cells. Thus, c-Myc drives Cul4b expression to inextricably link DNA damage response pathways to cell proliferation in CD8+ T cells (Supplementary Fig. 9i).

## Discussion

Here we demonstrated that c-Myc uses Cul4b to facilitate CD8+ T cell expansion and anti-viral memory and effector function. Following T cell activation, c-Myc associated with regulatory elements in the Cul4b gene to increase expression of Cul4b. c-Myc similarly regulated expression of other CRL4 components. By increasing levels of Cul4b, c-Myc connects DNA damage control pathways with cell proliferation. Inextricably linking these two processes ensures that rapidly dividing CD8+ T cells avoid replication stress as they expand to arm the host with effector and memory CD8+ T cells. Without Cul4b, the coordinated sequence of these events is disrupted, and attempts at proliferation result in a catastrophic collapse within 3–5 days of antigen encounter. Our data support that two key substrates, p21 and Cyclin E2, allow Cul4b to coordinate the pathways that regulate DNA damage response and cell cycle progression. Supporting the importance of Cul4b in CD8+ T cell anti-viral activity, mice lacking Cul4b in T cells were unable to clear virus (LCMV or Ectromelia), and after adoptive transfer, Cul4b-deficient LCMV-specific CD8+ T cells were unable to expand into memory or effector cells following virus infection. Together these data support a model in which c-Myc employs

Cul4b to allow for the rapid expansion, differentiation and function of memory and effector CD8+ T cells.

c-Myc is a multifunctional transcriptional factor that regulates cell cycle, growth, metabolism differentiation, and apoptosis by coordinately organizing multiple biosynthetic, bioenergetic and epigenetic processes in T cells[11,14,30]. In transformed cell lines, c-Myc has been shown to drive cell proliferation by coordinately regulating the DNA damage response needed during states of high replication stress[39]. However, whether and how c-Myc enables these processes under physiological conditions, was unclear. Here we demonstrated that c-Myc was required for the increased expression of Cul4b that followed TCR activation. We found c-Myc bound to Cul4b gene regulatory elements and c-Myc-deficient CD8+ T cells failed to increase Cul4b following TCR stimulation. Interestingly, c-Myc- and Cul4b-deficient CD8+ T cells showed significant overlap in their dysregulated pathways, including the DDR and pathways associated with cell proliferation. Supporting a role for Cul4b in these pathways, we determined that Cul4b reduced replication stress in activated T cells and thus allowed CD8+ T cell proliferation. This supported a specific role for Cul4b during DNA replication.

DNA damage accumulates during DNA synthesis when there is an increased number of active origins of replication and during scenarios of rapid S-phase progression, as occurs in expanding CD8+ T cells[3,40,41]. To ensure replication happens only once during the cell cycle, the two steps of replication initiation, origin licensing and firing, are temporally separated. This separation is coordinated by changes in CDK activity. Specifically, high CDK activity triggers origin firing and marks the start of S phase[42]. Cdk2 phosphorylates Type E Cyclins (Cyclin E1 and E2). These Cyclins accumulate at the G1/S transition, where they promote entry into S phase and other DNA replication–associated functions[43,44]. In mammalian cells, Cyclin E levels specifically decline during S phase, reaching low or undetectable levels by the time replication is complete[45]. This decline of Cyclin E1 and E2 levels is due to ubiquitin-mediated degradation, and when levels are not reduced, there is an increased risk of genomic instability[46]. In transformed cell lines, Cul4b was shown to regulate turnover of Cyclin E1 during S phase[47]. However, it was not clear whether Cul4b regulation was unique to Cyclin E1 or whether it might also regulate cyclin E2. Our data support that in CD8+ T cells, Cul4b promotes the degradation of Cyclin E2. Supporting this, Cul4b-deficient CD8+ T cells showed increased levels of Cyclin E2 associated with the DNA following DNA damage. Additionally, Cul4b-deficient CD8+ T cells contained increased levels of Cyclin E2 when assessed

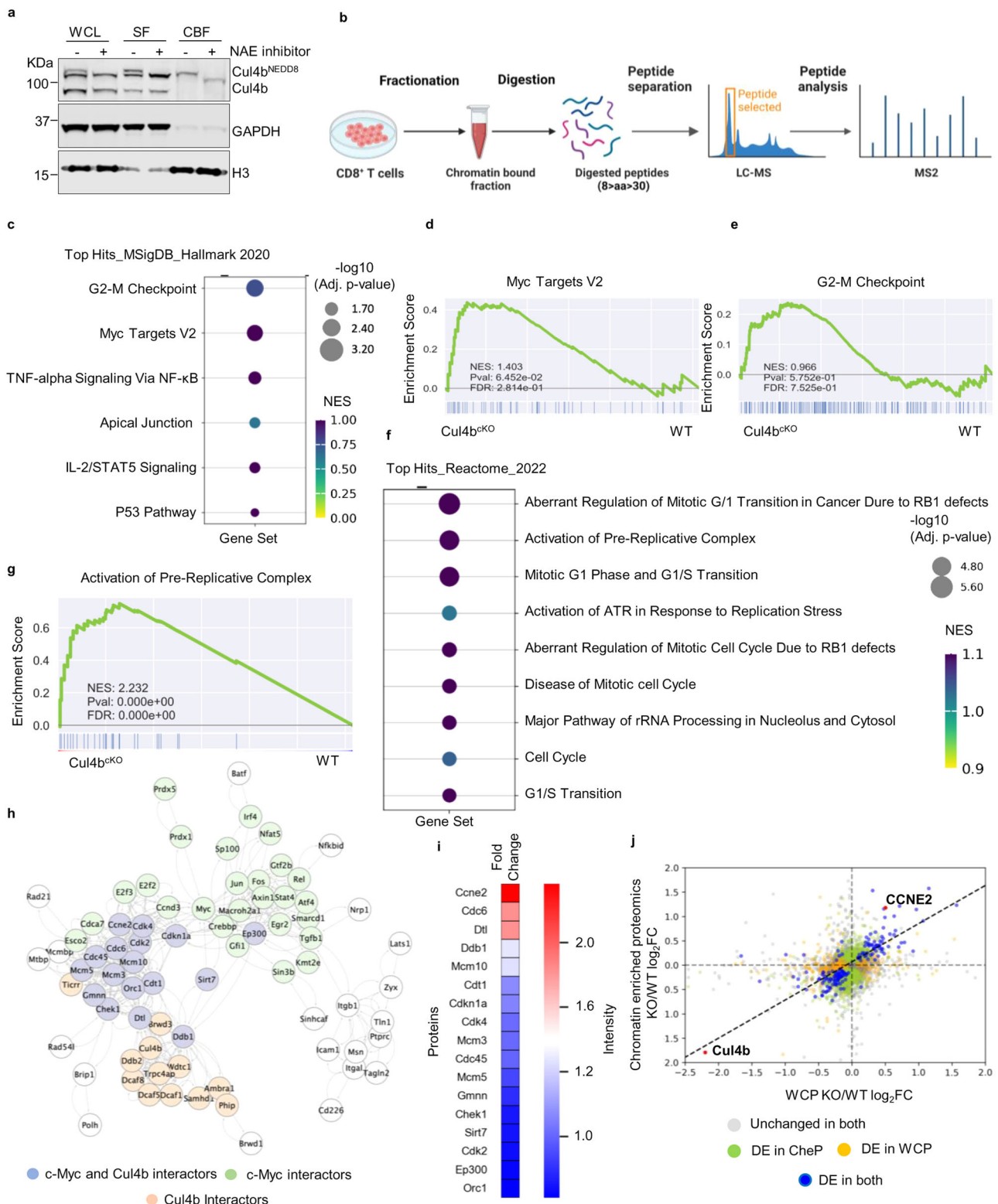

using two different methods. We found higher Cyclin E2 in Cul4b-deficient CD8+ T cells by whole cell proteomics and when we assessed the chromatin bound fraction using both immunoblot and mass spectrometry. In contrast, RNA levels were not increased in Cul4b-deficient cells. Bortezomib treatment of WT cells caused Cyclin E2 to accumulate, supporting that it is degraded by the proteosome, and this treatment did not cause further accumulation in Cul4b-deficient CD8+ T cells. Together, these data support that Cyclin E2 is a substrate of Cul4b in CD8+ T cells. Interestingly, Cyclin E1 levels were

similar between Cul4b-deficient and control CD8+ T cells, supporting that in CD8+ T cells Cul4b preferentially degrades Cyclin E2. This is somewhat paradoxical given that Cyclin E1 and cyclin E2 share homology (46% identity) and have similar functional domains[48]. One possible explanation is that Cul4b uses distinct substrate receptors to degrade Cyclin E1 and cyclin E2 in different cell types. Further studies will be needed to determine which substrate receptors aid Cul4b in CD8+ T cells, and to identify those which are responsible for recruiting Cyclin E2 to the CRL4b complex.

**Fig. 6 | In CD8⁺ T cells, a loss of Cul4b disrupts cell cycle progression and the DNA damage response. a** Naïve CD8⁺ T cells were stimulated for 40 h and whole cell lysate (WCL), soluble fraction (SF), and chromatin-bound (CBF) proteins were fractionated and analyzed by immunoblot. Histone H3 and GAPDH were used to assess enrichment of chromatin bound fraction (CBF) and soluble faction (SF), respectively. Cul4b shows neddylated (slower migrating) and non-neddylated forms of the protein. Treatment of activated CD8⁺ T cells with Nedd8 activating enzyme (NAE) inhibitor (NAEi, 1 µM for last 1 h of culture) resulted in a loss of neddylated Cul4b. The data is the representative of three independent experiments. **b** Schematic of the Chromatin Enrichment for Proteomics in CD8⁺ T cells (ChEP) assay based on chromatin pellet isolation[70]. Schematic diagram was created with BioRender.com. **c** Hallmark gene sets were assessed among the differentially expressed proteins between control and Cul4b^cKO CD8⁺ T cells. G2-M checkpoint, c-Myc-targets are most significantly enriched in Cul4b^cKO CD8⁺ T cells. The size of the bubble represents the adjusted *p*-value while the green-blue color scale represents normalized gene enrichment scores. **d, e** GSEA on Hallmark gene sets of G2-M checkpoint and c-Myc-target proteins show their enrichment in Cul4b^cKO CD8⁺ T cells over control CD8⁺ T cells. **f** Reactome pathway analysis of differentially regulated proteins based on *p*-value and enrichment score. Notably, multiple terms related to cell cycle progression, activation of pre-replicative complex, checkpoints and DNA repair are among the top categories. **g** GSEA performed on the Reactome term "activation of pre-replicative complex" showed its enrichment in Cul4b^cKO CD8⁺ T cells over control CD8⁺ T cells. For (**c–g**) *p*-values represent the probability of overrepresentation via 2-tailed Fisher's exact tests and were corrected for multiple comparisons via the Benjamini-Hochberg procedure. Normalized enrichment score (NES) reflects the degree to which the gene set is overrepresented at the top or bottom of a ranked list of genes, normalized for gene set size and correlations between the gene set and expression dataset. **h** Protein–protein interaction network analysis. Networks, constructed by the STRING Protein–Protein Interaction Network (Ver 11.5) tool, revealed interactions among upregulated proteins identified in the Cul4b^cKO CD8⁺ T cells and how they clustered with c-Myc and Cul4b. Most of these proteins clustered together, and upon sub-clustering three nodes were identified. The green node depicts proteins within 2 degrees of interaction with c-Myc while those in orange are within 2 degrees of interaction with Cul4b. Proteins in blue were within 2 degrees of interaction with both Cul4b and c-Myc. **i** Heat map of 17 differentially upregulated proteins that were within 2 degrees of interaction with both Cul4b and c-Myc. Red indicates the most differentially regulated proteins within this group (eg. CCNE2). **j** ChEP enriched proteomics analysis increased the depth of the detection of chromatin associated proteins Scatter plot shows the correlation between whole cell proteome (WCP) and ChEP. Dots in gray indicate proteins unchanged in both datasets, orange indicates proteins differentially identified in WCP and blue indicates proteins differentially identified in both datasets. Green dots indicate proteins differentially identified in ChEP. Cul4b and CCNE2 (Cyclin E2) are marked in red for emphasis. Source data are provided as a "Source Data" file.

In CD8⁺ T cells, Cul4b deletion also resulted in the accumulation of Cdkn1 (p21), a CDK and cell cycle inhibitor. Several studies have shown that p21 can be ubiquitinated and degraded. Ligases implicated in this include the CRL4 complex[23]. Supporting that Cul4b promotes p21 degradation in CD8⁺ T cells, we found increased levels of chromatin bound p21 in Cul4b-deficient CD8⁺ T cells. Additionally, while p21 was not identified in the whole cell proteomics, we found increased levels of p21 in whole cell lysates and in the soluble fraction. However, these data are complicated by our finding that Cul4b deletion led to increased mRNA expression of p21. Thus, further studies are needed to confirm p21 is a substrate of Cul4b in CD8⁺ T cells. p21 function has been studied in T cells as well as in cancer cells and transformed cell lines. p21 can negatively regulate T cell proliferation following stimulation[18]. The induction of p21 by several growth factors and cytokines, including IFN-γ, has been reported[49]. p21 can inhibit cell cycle entry by blocking formation of active cyclin–CDK complexes or by inhibiting DNA replication[50]. In T cells, p21 overexpression limits the accumulation of effector/memory T cells, as well as their IFN-γ production[20]. When over-expressed, protracted accumulation of p21 resulted in an increase of replication licensing factors Cdt1, Cdc6 and Orc, resulting in attempts at re-replication and fork collapse, DNA damage and ensuing genomic instability if not repaired[36]. Our findings fit with these reports and support that high levels of p21 and Cyclin E2 in CD8⁺ T cells can result in similar events resulting in a loss of T cell expansion and effector/memory cell function. Thus, our data support a model in which Cul4b restrains replication stress likely by controlling levels of Cyclin E2 and p21.

The ability of c-Myc to promote cellular proliferation stems from its ability, as a transcription factor, to directly control the expression of a large number of cellular pathways implicated in S-phase progression, balancing replication stress and activation of the DNA damage response[8,39,51,52]. We found that c-Myc is required for the increased expression of proteins important in cell cycle progression and DNA damage response pathways. c-Myc also increases expression of Cul4b which then localizes to the chromatin to limit the accumulation of proteins implicated in replication stress. This suggests a more direct role of c-Myc in controlling S-phase progression possibly by directly participating in the licensing or assembly of (pre)-replicative complexes[53,54]. Previous studies have demonstrated that c-Myc directly regulates T cell proliferation through transcriptional control of cell cycle regulators or through metabolic reprogramming. Our studies advocate that c-Myc as an essential coordinator of cell cycle and DDR pathways during CD8⁺ T cell activation allowing for an extraordinarily rapid rate of cell proliferation. Given that antigen-activated lymphocytes are more prone to genomic stress due to their rapid rate of cell division, targeted manipulation of DNA damage-response (DDR) signaling pathways may allow for selective targeting of highly proliferative pathogenic T cells that drive disease.

In the spectrum of inflammatory illnesses, c-Myc dysfunction has been reported in patients with Crohn's disease[55] and other chronic gastrointestinal disorders[56]. In a model of autoimmune encephalomyelitis, treatment with c-Myc inhibitor 10058-F4 suppressed the ability of Th1-differentiated T-cells to induce inflammation[57]. To date, no specific drug is currently available to directly target c-Myc, mainly because of its "undruggable" properties. Most efforts for developing clinically applicable small molecule inhibitors of c-Myc have focused on targeting the Myc/MAX heterodimer[58], but the majority of compounds developed thus far show low cellular or in vivo potency and deleterious off-target effects. Data presented here could provide directions for the development of anti-inflammatory therapies. Specifically, targeting Cul4b could be a promising alternative.

## Methods
### Mice
C57BL/6 J (B6), B6.CD45.1, Rag1⁻/⁻, and CD4-Cre mice were from the Jackson Laboratory. The mice used were kept in a C57BL/6 J (B6) background. Cul4b^fl/fl mice was generated using CRISPR/Cas9 as described previously[25]. For some of the experiments, Cul4b^fl/fl mice were crossed to P14 TCR transgenic mice. Cul4b being X-linked gene, the male and female mice with conditional deletion of Cul4b were symbolized as Cul4b^cKO. All mice were bred in house under specific pathogen-free conditions in the animal facility at the Children's Hospital of Philadelphia (CHOP). The mice were housed at 18–23 °C with 40–60% humidity, with 12-h light and 12-h dark cycles. All mice, if not specifically mentioned in this manuscript, were 6–12 weeks of age, and both sexes were used without randomization or blinding. Animal housing, care, and experimental procedures were performed in compliance with the CHOP Institutional Animal Care and Use Committee. All procedures were conducted in accordance with the Animal Welfare Act and were approved by the Institutional Animal Care and Use Committee at Children's Hospital of Philadelphia (protocol ID 810).

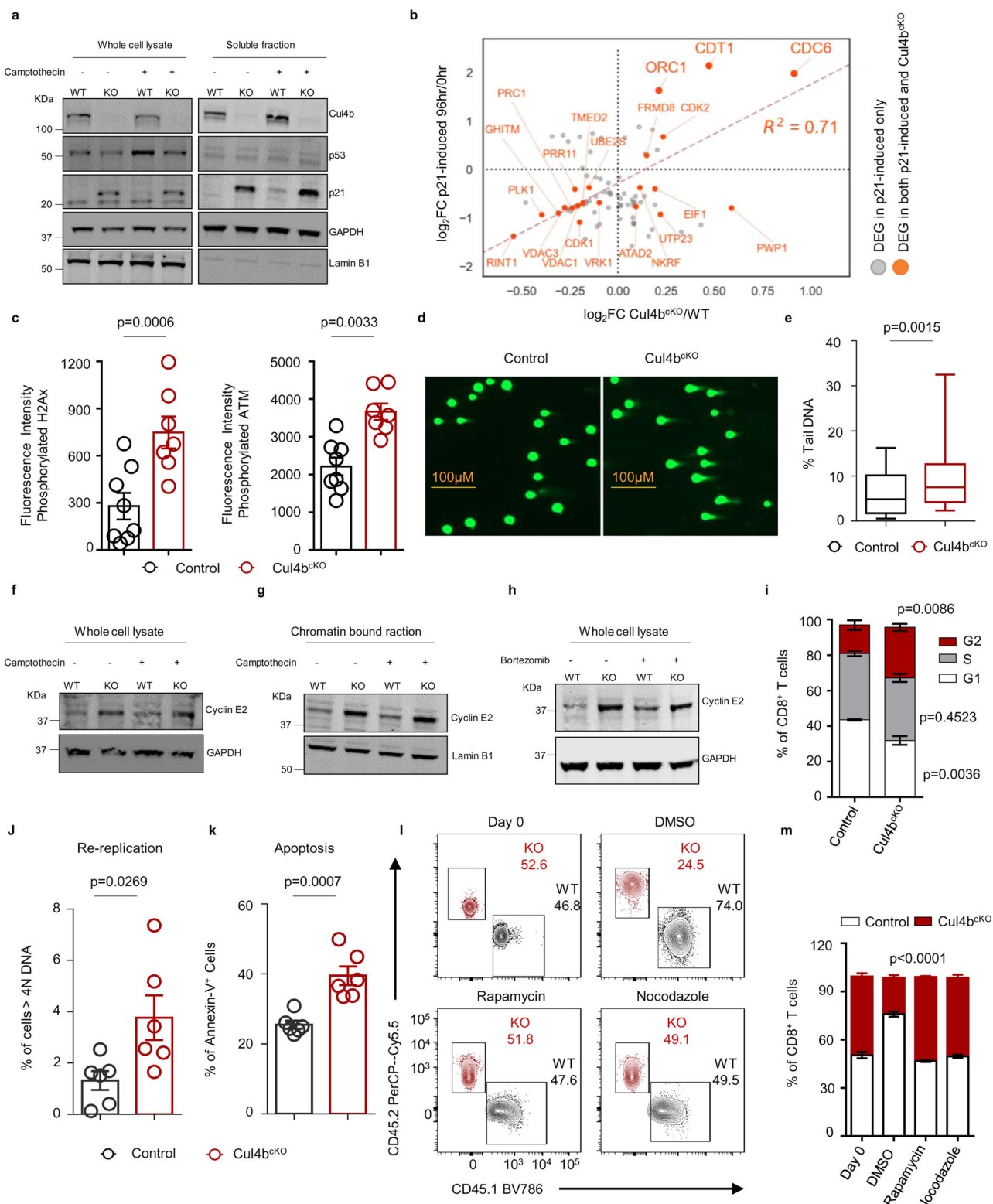

## LCMV and ectromelia infection

Mice were infected with LCMV Armstrong ($2 \times 10^5$ PFU per mouse) by intraperitoneal injection to establish acute infection. LCMV Armstrong were grown in BHK-21 cells, and Viral titers were determined by plaque assay using Vero cells.

Mice were inoculated with $1 \times 10^3$ pfu of ectromelia virus (ECTV) in the footpad and were monitored for limb swelling and loss.

## Bone marrow chimeras

For mixed bone marrow chimera (BMC) experiment, the recipient Rag1[-/-] mice were sublethally irradiated (400 rad) using an X-Rad Irradiator. Bone marrow (BM) cells from control (CD45.1) and Cul4b[fl/fl]-CD4Cre (CD45.2; Cul4b[cKO]) were depleted of T cells. Briefly, BM cells from donor mice was collected, RBCs were lysed and T cell depletion was done using Phycoerythrin (PE) conjugated TCR-β antibody (Clone H57-597 BioLegend) and anti-PE microbeads (Miltenyi Biotec). BM cells

**Fig. 7 | Cul4b circumvents replication induced stress in CD8⁺ T cells. a** Analysis of p21 levels in Cul4b deficient (Cul4bᶜᴷᴼ) and control CD8⁺ T cells, stimulated for 40 h with anti-CD3/CD28 mAbs (5 μg/mL) and treated with Camptothecin (2 μM) for the last 2 h. Whole cell lysate (WCL) and soluble fraction (SF), were analyzed by immunoblot. Lamin B1 and GAPDH were used as loading controls. Evaluation of p21 was changeling due to presence of the non-specific bands in the chromatin bound fraction. The data is the representative of three independent experiments. **b** Comparisons of proteins showing increased levels in p21 overexpressed cells with those elevated in Cul4bᶜᴷᴼ CD8⁺ T cells. Scatter plots showing log₂ fold change of upregulated proteins in p21 overexpressed cells (y-axis) versus proteins identified as elevated in Cul4bᶜᴷᴼ CD8⁺ T cells (x-axis). Proteins shared between the two data sets are shown in red; proteins unique to p21 overexpressing cells are in gray. **c** Activated CD8⁺ T cells from control and Cul4bᶜᴷᴼ mice infected with LCMV were analyzed for the expression of γ-H2AX, and pATM. Fluorescence minus one (FMO) controls were used to interpret the data. Data from $n = 8$ for control and $n = 7$ for Cul4bᶜᴷᴼ is shown as mean ± S.E.M. *p*-values were calculated using an unpaired two-tailed *t*-test. **d** Comet Assay to detect DNA damage in activated T cells treated with Camptothecin (CTP). CD8⁺ T cells were stimulated with anti-CD3/CD28 mAb. On day 2, cells were treated with Camptothecin (2 μM) for 1 h. **e** The percent tail DNA was calculated using OpenComet software. Data from $n = 2$ for control and $n = 3$ for Cul4bᶜᴷᴼ and from each sample $n > 80$ cells were analyzed. Box-whisker plot (displaying the 90/10 percentile at the whiskers and the median in the center line. Data is shown as mean ± S.E.M. *p*-values were calculated using an unpaired two-tailed *t*-test. **f, g** Cyclin E2 was analyzed in Cul4bᶜᴷᴼ and control CD8⁺ T cells that were

stimulated for 40 h with anti-CD3/CD28 mAbs (5 μg/mL) and were treated with Camptothecin (2 μM) for the last 2 h. The data is representative of three independent experiments. **h** Assessment of Bortezomib (BTZ) induced changes in Cyclin E2 by immunoblot. CD8⁺ T cells from control and Cul4bᶜᴷᴼ were treated with vehicle or BTZ (100 nM) for 4 h. β-actin was used as a loading control. The data is representative of three independent experiments. **i** CD8⁺ T cells were purified from control and Cul4bᶜᴷᴼ mice and stimulated with anti-CD3/CD28 mAbs (5 μg/mL) for 40 h. Cells were fixed and permeabilized, stained with PI, and DNA content was analyzed by flow cytometry. The bar graph shows the percentage of cells in, G1, S, and G2/M phases of the cell-cycle. Data from 4 independent experiments is shown as mean ± S.E.M. *p*-values were calculated using an unpaired two-tailed *t*-test. **j** Re-replication was assessed using PI staining. The graph illustrates the percentages of control and Cul4bᶜᴷᴼ CD8⁺ T cells with DNA > 4n. Data from $n = 6$ per genotype is shown as mean ± S.E.M. *p*-values were calculated using an unpaired two-tailed *t*-test. **k** Bar graph shows quantitation of the percentages of apoptotic (Annexin V positive) cells from three independent experiments. Data from $n = 6$ per genotype is shown as mean ± S.E.M. *p*-values were calculated using an unpaired two-tailed *t*-test. **l, m** Naïve CD8⁺ T cells isolated from control (CD45.1) and Cul4bᶜᴷᴼ (CD45.2) mice were co-cultured and stimulated in vitro with anti-CD3/CD28 mAbs (5 μg/mL) for 3 days in the presence of either DMSO, Rapamycin or Nocodazole as indicated. The relative proportions of cells were assessed by flow cytometry. Data is representative of two independent experiments Data from $n = 4$ per genotype is shown as mean ± S.E.M. *p*-values were calculated using a paired two-tailed *t*-test. Source data are provided as a "Source Data" file.

were Isolated and resuspended in MACS buffer. Stained with PE conjugated TCR-β antibody and anti-PE microbeads. After staining cell suspension was loaded onto the column to deplete BM cells of T cells, flow through was collected and used for the experiments. T cell depleted BM cells from donor mice were enumerated and mixed at a 1:1 ratio. Two million mixed BM cells in plain media were injected intravenously into recipient mice. The irradiated mice were kept on Sulfatrim antibiotic water and analyzed ≥8 wk after injection of BM cells.

## Adoptive transfer of T cells

For adoptive transfer experiment, naïve CD8⁺ T cells were isolated from spleens and lymph of naïve Cul4bᶜᴷᴼ or wild-type P14 mice by CD8 negative selection (Miltenyi Biotec). The indicated number ($5 \times 10^4$–$10^5$) of 1:1 mixed, or single P14 CD8⁺ T cells, were transferred by intravenous injection into nonirradiated naïve recipient mice (B6.CD45.1). For flow cytometry analysis after infection, major lymphoid organs were removed on the days indicated, and single cell suspensions were prepared as previously described[25]. RBCs in the cell suspensions were lysed using ammonium chloride.

## Flow cytometry and antibodies

Lymph nodes, spleen and lungs were mechanically dissociated and macerated through the 70-micron cell strainer. For lungs, single cell suspension was prepared using enzymatic digestion (Collagenase/DNase) at room temperature for 1 h. Single-cell suspensions were stained with a fixable viability dye, then pretreated with unlabeled anti-CD16/CD32 (Fc Block BD Pharmingen). Cells were then stained in FACS buffer (PBS containing 2.5% fetal calf serum and 0.1% sodium azide) with mixtures of directly conjugated antibodies. Following antibodies TCRβ [Clone H57-597,catalog numbers 109208 and109205;1:300], CD3 [Clone 17A2, 145-2C11, catalog numbers 100237and 100328;1:300], CD4 [Clone GK1.5 and RM4-5, catalog numbers 100545, 100406, 100510, 100434,100422;1:300], CD8a [Clone 53-6.7, catalog numbers 100743, 100738, 100751, 100706, 100714, 100722, 100712, 100730;1:300], CD19 [Clone 6D5 catalog numbers 115543, 115520;1:300], CD44 [Clone IM7, catalog numbers 103020, 103006, 103056, 103026, 103027, 103012;1:300], CD45.1 [Clone A20, catalog numbers 110713, 110716, 110728, 110737, 110731; 1:300], CD45.2 [Clone 104, catalog numbers 109824, 109830, 109820, 109828;1:300], B220 [Clone RA3-6B2, catalog number

103206;1:300], CD62L [Clone MEL-14 catalog number 104448, 104428; 1:300], CD90.2 [Clone 30-H12 catalog numbers 105306, 105319, 105327; 1:300], CD107a [Clone 1D4B, catalog number 121611; 1:100], CD127 [Clone A7R34, catalog number 135039; 1:200], KLRG1 [Clone 2F1, catalog number 138429; 1:300], TNF-α [clone MP6-XT22, catalog numbers 506308 and 506318; 1:200], IFN-γ [Clone XMG1.2, catalog number 505826; 1:200], ATM phospho Ser1981 [Clone 10H11.E12, catalog number 651203; 1:100], H2A.X phospho Ser139 [Clone 2F3, catalog number 613407; 1:100], CD62L (Clone MEL-14, catalog number 104448, 104428; 1:200], I, Ki-67 [1Clone 6A8 catalog number 652411;1:200] from BioLegend. TCR Vβ Screening Panel catalog number 557004 (Vβ2 [B20.6;1:100], Vβ4 [KT4:1:100], Vβ5 [MR9-4;1:100], Vβ6 [RR4-7;1:100], Vβ7 [TR310;1:100], Vβ8 [MR5-2;1:100] and Vβ17 [KJ23;1:100]) from BD Bioscience. PD-1 [Clone J43 catalog number 11-9985-81; 1:300], KLRG1 [Clone 2F1, catalog numbers 46-5893-80, 25-5893-80, 17-5893-81;1:300] CD127 [A7R34, catalog numbers 17-1271-82 and 47-1271-80; 1:200] from eBiosciences. Annexin V [Lot number 2354351, catalog number 640906] from invitrogen MHC class I H-2Db gp33 tetramers (1:200) and MHC class II I-Ab gp66 tetramers (1:200) were obtained from the National Institute of Health Tetramer Core.

For intracellular cytokine staining, single cell suspension was incubated with or without 50 ng/ml gp33 peptide or PMA Ionomycin in the presence of GolgiPlug and GolgiStop for 4 h at 37 °C and stained using the Cytofix/Cytoperm kit (BD). For intracellular transcription factor staining, Foxp3 staining kit (eBioscience) was used according to manufacturer's instruction. For the T cell proliferation assay, naïve CD8⁺ T cell were isolated from control (CD45.1) and Cul4bᶜᴷᴼ(CD45.2) and were cocultured for 3 or 5 days. Briefly, CD8⁺ T cells (100,000/well) were incubated with Cell Trace Violet (CTV) (Thermo Fisher Scientific) at a final concentration of 5 μM for 10 min at 37 °C. Cells were washed three times with ice-cold complete RPMI 1640 and stimulated with anti-CD3 [145-2C11; BioLegend]/CD28 [37.51; BioLegend] antibodies (5 μg/mL). After day 3 cells were harvested and change in CTV intensity was measured. The relative percentages of cells in co-culture were analyzed at both day 3 and day 5. Samples were analyzed using a Fortessa (BD Biosciences) flow cytometer at the CHOP Flow Cytometry Core Facility. Data was analyzed using FlowJo software V10 (TreeStar Ashland). Results are expressed as the percentage of positive cells or median fluorescence intensity (MFI).

## Naïve CD8+ T cell isolation and in vitro stimulation

Naïve CD8+ T cells were isolated by magnetic separation using the Miltenyi Naïve CD8+ T cell isolation kit. Briefly, cells were isolated from spleen and lymph nodes. Single cell suspension was resuspended in MACS buffer and stained with antibody conjugated beads. For naïve CD8+ T cells isolation, a cocktail of biotin-conjugated monoclonal antibodies against CD4, CD11b, CD11c, CD19, CD25, CD45R (B220), CD49b (DX5), CD105, Anti-MHC class II, Ter-119, and TCRγ/δ was used. Then microbeads conjugated to monoclonal anti-biotin antibodies and CD44 microbeads were added. After staining, cell suspension was loaded onto the column and cells that flow-through the column (unlabeled cells) were the enriched naïve CD8+ T cells.

Naïve CD8+ T cells were stimulated in vitro in complete RPMI (RPMI 1640 supplemented with 10% fetal bovine serum (Atlanta Biologicals), HEPES (Thermo Fisher Scientific), nonessential amino acids, sodium pyruvate (Thermo Fisher Scientific), 2 mM-Glutamine, antibiotics and 2-mercaptoethanol) with plate bound anti-CD3 (Clone 17A2 BioLegend) and anti-CD28 (Clone 37.51 BioLegend) antibodies. The tissue culture plates were coated with the antibodies overnight at 4 °C (5 μg/mL). Cells were cultured at 37 °C with 10% $CO_2$. For western blotting experiments prior to harvesting, cells cultures were supplemented with DMSO or Bortezomib (50 nM) for 2–4 h.

## Cell cycle analysis

For cell cycle analysis, CD8+ T cells were incubated in the 24-well culture plate with α-CD3/CD28 mAb (5 μg/mL) for 48 h to generate asynchronous population. After 48 h, cells were harvested and fixed by adding chilled 70% ethanol dropwise under constant vortexing. Cells were washed and stained with FxCycle™ PI/RNase Staining Solution (Thermo Fisher Scientific) as per manual instructions. Cells were incubated at room temperature for 30 min and cell cycle was assessed on Fortessa (BD Biosciences) flow cytometer and analyzed using FlowJo software V10 (TreeStar Ashland).

## Comet assay

Neutral comet assay (Abcam) was used to detect DSBs and DNA lesions as per manufactures instructions with slight modifications. Briefly, CD8+ T cells were stimulated with anti-CD3/CD28 mAb for 40 h and for the last 1 h in culture Camptothecin (Sigma-Aldrich) (ETP; 20 μM) was added to induce DNA damage. Cells were harvested and mixed with low melting temperature agarose at 1/10 ratio (v/v) and layered on slides pre-coated with one layer of agarose. Slides were lysed in lysis buffer (containing NaCl, EDTA, 1% DMSO) overnight at 4 °C and then subjected to electrophoresis in pre-chilled TBE electrophoresis buffer at 1 Volt/cm for 15 min. Slides were washed with pre-chilled DI $H_2O$ and 70% ethanol and then air dried. Slides were stained with Vista Green DNA Dye for 15 mins and visualized under Evos FL Auto epi-fluorescence microscopy (Thermo Fisher Scientific). Analysis was performed with OpenComet softwarev1.3.1. Tail DNA % was measured to show the extent of DNA damage. At least 80 cells per sample were analyzed.

## Western blotting

Cells were washed with phosphate-buffered saline (PBS) ($Ca^{2+}$ and $Mg^{2+}$-free) and lysed in SDS sample buffer (62.5 mM Tris-HCl, 2% w/v SDS, 10% glycerol, 50 mM DTT, 0.1% bromophenol blue, Protease inhibitor cocktail (Roche), Halt phosphatase inhibitor cocktail (Thermo Fisher Scientific), $Zn^{2+}$-chelator *ortho-phenanthroline* (o-PA) (LifeSensors), deubiquitinase inhibitor PR-619 (LifeSensors). Samples were sonicated to reduce viscosity, denatured by boiling, and then cooled on ice. Samples were resolved on 4–12% Novex Tris-Glycine Gels (Thermo Fisher Scientific) and then transferred onto PVDF membrane (Amersham Pharmacia Biotech, Piscataway, NJ). The membrane was probed with the primary antibodies Cul4b (Sigma Aldrich; lot number B10150; catalog number HPA011880 1:500), Cul4b

(ProteinTech, catalog number 12916-1-AP; 1:1000), p53 [Abcam, clone pab122, catalog number ab90363; 1:1000), Lamin B1 (Cell Signaling Technology, Clone D4Q4Z, catalog number 12586 S; 1:1000), c-Myc (Cell Signaling Technology, Clone D84C12, catalog number 5605 S; 1:1000), Cyclin E1 (Santa Cruz; Clone M-20, catalog number sc-4811:500; 1:500), Cyclin E2 (Cell Signaling Technology, Lot number 3, catalog number 4132 S; 1:1000), p21 (Santa Cruz; clone F-5, catalog number sc-6246; 1:200), GAPDH (Millipore, Lot number 2910381, catalog number MAB374; 1:5000) and β-actin (Santa Cruz, Clone C4, catalog number, sc-47778; 1:5000) were used. Immunostaining was performed using appropriate goat anti-mouse (Invitrogen, lot number XA336885, catalog number A32730; 1:5000) and goat anti-rabbit (Invitrogen, lot number2260898 catalog number A21109) secondary antibodies and developed with LICOR Odyssey imaging system.

## Subcellular fractionation

Subcellular fractionation was carried out as described[59] with some modifications. α-CD3/CD28 mAb stimulated CD8+ T cells (40 h) in presence or absence of Camptothecin (2 μM) were harvested and washed twice with phosphate-buffered saline (PBS) ($Ca^{2+}$ and $Mg^{2+}$-free) and resuspended in solution A (10 mM HEPES, pH 7.4, 10 mM KCl, 1.5 mM $MgCl_2$, 0.34 M sucrose, 10% glycerol, 1 mM dithiothreitol, 10 mM NaF, 1 mM $Na_2VO_3$, EDTA free complete protease inhibitor cocktail (Roche), deubiquitylase inhibitors PR-619 (LifeSensors) and $Zn^{2+}$-chelator *ortho-phenanthroline* (o-PA) (LifeSensors). Triton X-100 was added to a final concentration of 0.1%, and the cells were incubated for 30 min on ice. Soluble fraction (SF) was separated from chromatin bound fraction by centrifugation (4 min, 1800 × g). Chromatin bound fraction was washed once in solution A and final chromatin pellet was resuspended in SDS sample buffer and sonicated to release chromatin-bound proteins.

## Sample preparation for whole cell and chromatin enriched proteome

Naïve CD8+ T cells were isolated by MACS from control and Cul4b$^{cKO}$ mice. Cells were stimulated for 40 h with anti-CD3/CD28 mAbs. After stimulation cells were harvested and cell pellets were stored at −80 °C. For Chromatin enriched proteomics (ChEP) after 40 h stimulation cells were treated with Camptothecin for 2 h. The chromatin bound fraction was isolated as described above and enriched chromatin was stored at −80 °C.

**Protein extraction.** Samples were solubilized in 50 μL of extraction buffer containing 5% sodium dodecyl sulfate (SDS, Affymetrix), 50 mM TEAB (pH 8.5, Sigma), and protease inhibitor cocktail (Roche Complete, EDTA free). To shear DNA and ensure complete solubilization, samples were sonicated for 10 min at 10 °C in a Covaris R230 focused-ultrasonicator with the following settings: Dithering: $Y = 3.0$, Speed = 20.0, PIP: 360.0, DF: 30, CPB: 200. Samples were centrifuged at 3000 g for 10 min to clarify lysate. 1 μL of each sample was taken to estimate protein concentration by in-gel staining with Bradford Coomassie solution and intensity analysis with GelAnalyzer 19.1, using a serial dilution of an in-house generated E.coli lysate standard.

**In-solution digestion.** 100ug of each sample was digested per the S-Trap Micro (Protifi) manufacturer's protocol[60]. Briefly, proteins were reduced in 5 mM TCEP (Thermo), alkylated in 20 mM iodoacetamide (Sigma), then acidified with phosphoric acid (Aldrich) to a final concentration of 1.2%. Samples were diluted with 90% methanol (Fisher) in 100 mM TEAB, then loaded onto an S-trap column and washed three times with 50/50 chloroform/methanol (Fisher) followed by three washes of 90% methanol in 100 mM TEAB. A 1:10 ratio (enzyme: protein) of Trypsin (Promega) and LysC (Wako) suspended in 20 μL 50 mM TEAB was added, and samples were digested for 1.5 h at 47 °C in a humidity chamber. After incubation, peptides were eluted with an

additional 40 μL of 50 mM TEAB, followed by 40 μL of 0.1% tri-fluoroacetic acid (TFA) (Pierce) in water, and finally 40 μL of 50/50 acetonitrile:water (Fisher) in 0.1% TFA. Eluates were combined and organic solvent was dried off via vacuum centrifugation. Samples were then desalted using an Oasis HLB μElution plate (30um, Waters). Wells were conditioned two times with 200 μL of acetonitrile and equilibrated three times with 200 μL of 0.1% TFA. Samples were applied, washed three times with 200 μL 0.1% TFA, and eluted directly into autosampler vials in three increments of 65 μL of 50:50 acetonitrile:water. Eluates were then dried by vacuum centrifugation and reconstituted in 0.1% TFA containing iRT peptides (Biognosys, Schlieren, Switzerland). Peptides were quantified with A280 measurement on a NanoDrop 1000 (Thermo) and adjusted to 0.4 μg/μL for injection.

**Generation of spectral library.** The spectral library was generated by pooling peptides from each sample. This mix was lyophilized, solubilized in 2% acetonitrile with 5 mM ammonium formate pH 10 (Sigma 17843), and separated by high pH RP-HPLC into 72 fractions which were recombined in a concatenated fashion into 12 sub-fractions[61,62]. These were dried by vacuum centrifugation and solubilized in 0.1% TFA containing iRT peptides (Biognosys AG, iRT).

**Mass spectrometry data acquisition.** Samples were randomized and analyzed on an Exploris 480 mass spectrometer (Thermofisher Scientific San Jose, CA) coupled with an Ultimate 3000 nano UPLC system and an EasySpray source. Peptides were loaded onto an Acclaim PepMap 100 75 um × 2 cm trap column (Thermo) at 5uL/min, and separated by reverse phase (RP)-HPLC on a nanocapillary column, 75 μm id × 50 cm 2 um PepMap RSLC C18 column (Thermo). Mobile phase A consisted of 0.1% formic acid and mobile phase B of 0.1% formic acid/acetonitrile. Peptides were eluted into the mass spectrometer at 300 nL/min with each RP-LC run comprising a 90-min gradient from 3% B to 45% B. For data dependent acquisition (DDA), the mass spectrometer was set with a master scan at $R = 120000$, with a scan range of 300–1400, and standard AGC target. Maximum injection time was set to auto, and dynamic exclusion was set to 30 s for one repeat. Charge state 2–5 were included. Top 15 data dependent MSMS scans were collected at $R = 45000$, with first mass at 120, normalized AGC target at 300%, automatic maximum injection time, and HCD NCE at 30. For data independent acquisition (DIA) mass spectrometer settings were as follows: one full MS scan at 120,000 resolution with a scan range of 350–1200 m/z and normalized automatic gain control (AGC) target of 300%, and automatic maximum inject time. This was followed by variable (DIA) isolation windows, MS2 scans at 30,000 resolution, a normalized AGC target of 1000%, and automatic injection time. The default charge state was 3, the first mass was fixed at 250 m/z, and the normalized collision energy for each window was set at 27.

**System suitability and quality control.** The suitability of Exploris 480 instrument was monitored using QuiC software (Biognosys, Schlieren, Switzerland) for the analysis of the spiked-in iRT peptides. Meanwhile, as a measure for quality control, we injected standard E. coli protein digest before in the middle of, and after sample set using DDA mode. The collected DDA data were analyzed in MaxQuant[63] and the output was subsequently visualized using the PTXQC[64] package to track the quality of the instrumentation.

**Database searching.** The raw files for spectral library were processed with Spectronaut version 16.2[65] using reference mouse proteome from Uniprot (25,436 reviewed canonical and isoform proteins) and the MaxQuant list of protein contaminants (245 proteins). The default settings in Spectronaut were used for spectral library generation with cysteine carbamidomethylation as fixed modification and methionine oxidation and protein N-terminal Acetylation as variable modifications.

The generated spectral library was composed of 191,028 precursors (8310 protein groups) and was employed to process the individual DIA files. Protein MS2 intensity values were measured, and cross-run normalized by Spectronaut. Data were filtered out at a false discovery rate (FDR) of 1% at precursor, peptide and protein level.

The DDA files were processed with MaxQuant (2.0.3.0) using a mouse reference database from Uniprot appended with sequences for protein contaminants. Carbamidomethyl of Cys was defined as a fixed modification. Oxidation of Met and Acetylation of protein N-terminal were set as variable modifications. Trypsin/P was selected as the digestion enzyme. The False Discovery Rate for precursors, peptides, and proteins were set at 1%. Fragment ion tolerance was set to 0.5 Da. The MS/MS tolerance was set at 20 ppm. The minimum peptide length was set at 7 amino acids. The rest of the parameters were kept as default.

### Protein-Protein Interaction Network (PPIN)

Scored protein interaction data in *Mus musculus* were downloaded from the STRING database (10090, protein.links.v11.5) and filtered for a combined interaction score of 0.750 or above (out of 1.000 maximum). Of the differentially regulated proteins in the Myc[cKO] CD8[+] T cell dataset (PXD016105)[30], proteins overlapping with the GOBP categories of cell cycle, DNA repair, or cellular response to DNA damage were superimposed with the scored protein interaction data from STRING and visualized in Cytoscape (v. 3.9.1); this generated a network of 126 nodes and 1562 interaction edges. A similar method was used for the differentially upregulated proteins in the Cul4b[cKO] CD8[+] T cell chromatin-enriched dataset, generating a network of 148 nodes and 1908 interaction edges. Of those, 76 nodes were spatially clustered with either Myc or Cul4b. 10 genes were within 1-degree of interaction with Cul4b, and 28 proteins were within 2-degrees of interaction with Cul4b. Eleven proteins were within 1-degree of interaction with Myc, and 42 proteins were within 2-degrees of interaction with Myc. Seventeen proteins were within 2 degrees of interaction with both Cul4b and Myc, including Cul4b associated proteins DTL and DDB1, as well CCNE2 (Cyclin E2), CDK2, CDC6, CDT1, and p21 (Cdkn1a). Nodes were spatially positioned using Cytoscape's edge-weighted spring embedded layout. Nodes with no interactions with other nodes in the network were filtered out.

### Correlation plots

Correlation plots for the p21 overexpression dataset (PXD004140)[36] and Cul4b ChEP dataset. There were 62 proteins that were found differentially expressed in p21 overexpressed dataset. Differential expression was defined by comparing 96 h vs 0 h post p21 overexpression. The differentially expressed p21 proteins, were compared to differentially expressed proteins in Cul4b[cKO] ChEP dataset (Student's *t*-test, alpha < 0.05). Log2fold-changes (log2FC) were obtained for both datasets for plotting. For the proteins differentially expressed in both datasets, a least-squares linear regression line of best fit and its associated Pearson correlation coefficient were calculated using their respective log2FC values.

Correlation plots for the Cul4b proteomics datasets were created using a concatenated dataset of proteins found in both the Cul4b[cKO] WCP and ChEP datasets; there were 5378 proteins found in both datasets, which were used for plotting. For each dataset, log2fold-changes (log2FC) of each protein were obtained by comparing the Cul4b[cKO] protein intensities to the control (WT) protein intensities. Student's *t*-tests were performed using alpha <0.05 for DE protein identification. A least-squares linear regression line of best fit and its associated Pearson correlation coefficient were calculated for the proteins that were differentially expressed in both datasets.

## Bioinformatics analysis

Differentially expressed proteins were analyzed for enrichment in numerous pathway analysis datasets: Gene Ontology for Biological Processes (m5.go.bp.v2022.1), Reactome (v.2022), and MSigDb Hallmark Genes (v.2020). Top GOBP and Reactome terms sorted by Bonferroni-corrected *p*-values were found using the Functional Annotation tools in Database for Annotation, Visualization and Integrated Discovery (DAVID). Top terms and associated adjusted *p*-values for MSigDb Hallmark Genes were found using the Enrichr API function in the gseapy package (v.1.0.4) in Python. When applicable, z-scores were obtained by finding the proportion of up- vs. downregulated genes in the respective gene ontology group; positive z-scores indicate that more differentially expressed proteins in the gene ontology group are upregulated than downregulated, while negative z-scores indicate the more proteins in that group are downregulated than upregulated. Visualization of gene ontology was performed using modified functions from the GOPlot library (https://wencke.github.io/) in R, modified functions from the gseapy package in Python, and the ClueGO application in Cytoscape (v. 3.9.1) for GOBP network visualization.

## RNA-seq and data analysis

Raw count data for control (WT) and Myc^cKO (KO) CD8+ T cells activated with anti-CD3/28 mAbs and ICAM for 36 h were pulled from GSE183619[14]. While for control (WT) and Cul4b^cKO (KO) CD8+ T cells, Naïve cells were isolated by MACS from control and Cul4b^cKO mice. Cells were stimulated for 24 h with anti-CD3/CD28 mAbs and total RNA was isolated using TRIzol reagent (Thermo Fisher Scientific) and poly-A selection was used to remove ribosomal RNA. After first and second strand synthesis from a template of poly-A selected/fragmented RNA, other procedures from end-repair to PCR amplification were performed. The DNA library was quantitated using Qubit. Libraries were sequenced on BGIseq500 platform with 50 bp single-end sequencing. Single-end sequencing reads were quality checked using FASTQC (www.bioinformatics.babraham.ac.uk/projects/fastqc), and Reads were aligned to the Ensembl *Mus musculus* reference genome (GRCm38) using the spliced transcripts alignment to a reference (STAR) alignment program[66]. Transcripts per million (TPM) were calculated from the raw read counts by obtaining gene lengths using MyGene in Python and calculating reads per kilobase (RPK) for each gene. The RPK values were summed per sample and divided by 1,000,000 to obtain each sample's "per million" scaling factor; RPK divided by the scaling factor outputted TPM. Student's *t*-tests were performed between WT and KO samples, and log₂-fold changes (log₂FC) of KO expression over WT expression were calculated for each gene. For Myc^cKO CD8+ T cells, differentially expressed genes were defined as genes with KO/WT log₂FC > 2 or KO/WT log₂FC < −2, and *p*-value < 0.001 between WT and KO. For Cul4b^cKO CD8 + T cells, DEGs were defined as genes with KO/WT log₂FC > 0.20 and *p*-value < 0.05.

## ChipSeq analysis

CD8+ T cell anti-c-Myc and input ChIP-seq BedGraph data mapped to the mm9 genome were obtained from GSE58081[67] and visualized in Integrated Genome Browser (IGB; v.9.1.6). Homer peak calling for c-Myc was performed by the original authors using a *p*-value cut of 5e-9. Genes of interest and associated intron and exon locations were loaded in IGB using RefSeq, and normalized peaks were compared between anti-Myc and input ChIP-seq samples. Vertical axes were equalized to the same value for both anti-Myc and input samples for visualization

## scRNA-seq analysis

The corresponding cell-gene UMI tables for day 32, 60 and 90 post-LCMV-infection in splenic CD8+ T cells were obtained from GSE131847[68] and imported, tagged, and concatenated into an AnnData object in Python (v. 3.9.13) using the AnnData package

(v. 0.8.0). Subsequent analysis was performed using the ScanPy package (v. 1.9.1) in Python. Cells expressing <200 genes and genes present in <3 cells were removed. Additionally, cells expressing >6100 genes, or cells with >7% mitochondrial genes, were removed. For clustering purposes, highly variable genes, identified as genes with high dispersion, were isolated. Next, we regressed out effects of total counts per cell and the percentage of mitochondrial genes expressed. Each gene was scaled to unit variance, and values exceeding a standard deviation of 10 were clipped. For clustering, principal component analysis (PCA) was performed using SVD solver ARPACK to obtain a Scree plot, which was used to identify the number of PCs for efficient clustering. For temporal analysis, violin plots for each time point were generated for all cells with at least one transcript per 10,000 counts. For memory subset analysis, similar biomarker criteria as by the original authors was used to classify the cells as long-lived effector (LLE), central-memory (T_CM), or effector-memory (T_EM). CD8+ T cells from day 32 post-infection were used for memory subset analysis for Cul4b-associated proteins.

## Statistical analysis

Results were expressed as the mean ± standard error of mean (S.E.M). Statistical analysis was performed using Prism software 7 and 9. *p*-value was calculated using the student two tailed *t*-test or Mann–Whitney *U*-test or Fisher's exact test. Two-sided *p*-values < 0.05 were considered statistically significant.

## Reporting summary

Further information on research design is available in the Nature Portfolio Reporting Summary linked to this article.

## Data availability

The mass spectrometry proteomics data has been deposited to the ProteomeXchange Consortium via the PRIDE[69] partner repository with accession number PXD041220. The bulk-RNA-seq data have been deposited in the GEO repository under accession codes GSE228695. The publicly available datasets with accession numbers GSE131847, GSE58081, GSE183619, PXD004140, PXD016105, PXD012058 were used in this study. All other data supporting the findings are available within this paper, and supplementary information. Source data are provided with this paper.

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

## Acknowledgements
We thank the Proteomics Core (RRID:SCR_023099), Flow Cytometry Core and Department of Veterinary Resources at the Children's Hospital of Philadelphia for their technical support. We thank the NIH Tetramer Core Facility for the tetramers. No specific funding was received for this work.

## Author contributions
Conceptualization: A.D. and P.O. with some input from SG, LE and EB. Methodology: A.D., S.M.G., K.K., S.R., I.G., K.S.F., N.P., Y.O., J.R., D.D.K., H.F. and L.S. Investigation: A.D., S.M.G., K.K., S.R., I.G., K.S.F., N.P., Y.O., J.R. and L.S. Visualization: A.D., D.D.K., P.O., S.M.G. Funding acquisition: P.O. Project administration: N.P. Supervision: P.O., A.D. and E.M.B. Writing—original draft—A.D. with some input from P.O. and D.D.K. Writing—review and editing: All authors.

## Competing interests
The authors declare no competing interests.
