## [Peer Review File · Nature Communications]

REVIEWER COMMENTS

Reviewer #1 (expert in T cell memory):

This manuscript builds upon previous work from the Oliver lab investigating the role of Cul4b regulation of DNA damage response in CD4 T cells. Here the authors identify a role for Cul4b in CD8 T cell proliferation, effector response, and memory formation. Through transcriptomic, proteomic, and cellular immunologic methods, the authors identify a regulatory network where c-Myc upregulates Cul4b in T cells to mitigate activation-induced DNA damage and cell cycle entrance through regulation of p21 and Cyclin E2.

This manuscript provides insight into the molecular mechanisms of managing T cell replicative stress. While this manuscript provides new insight into the interactions between c-Myc, Cul4b, p21, and Cyclin E2, there are a few concerns regarding the model and molecular mechanisms of these interactions.

Major Comments:

- Data supporting normal thymic development of the Cul4b-cKO animals would be important to demonstrate that the Cul4b naïve T cells are comparable to WT cells prior to activation. Enumeration of thymic developmental stages as well as repertoire analysis would be beneficial. In the Dar AA 2021 PLoS Biology publication, there were data to suggest decreased CD8 T cells in the thymus and spleen of Cul4b-cKO mice which may impact the interpretations of the peripheral T cell responses examined within this manuscript. An inducible knock-out system may complement or improve the design of these experiments to determine the contribution of Cul4b in peripheral T cell responses and remove any confounding variables from the proliferative burst during thymic development.
- The authors investigate the impairment of the DNA damage response in Cul4b-cKO cells but, it would be helpful to use an additional functional readout like a Comet assay that was used in the Dar AA et al paper. The addition of this assay would complement other data in Figure 7 of this manuscript to strengthen the evidence of increased DNA damage in Cul4b-cKO CD8 T cells.

Minor Comments:

- The authors propose an interactive model between c-Myc, Cul4b, p21, and Cyclin E2. It would be beneficial to provide a summary diagram that highlights these interactions in the WT and KO conditions.
- Additional citations within the introduction and results would be helpful in lines 57, 73, 77, 107.
- A description of the c-Myc conditional knock-out mouse model is missing from the methods section.
- In line 306, the marker is listed as 'CXC3CR1' instead of 'CX3CR1'.
- In line 322, the authors state that the number of P14 cells is decreased but this should state frequency since enumeration of cells was not shown for this data.
- The figures would benefit from using open/closed circles or another strategy to distinguish between experimental groups in graphs. When printed in black and white, it is very difficult to tell which group is which in the figures (see Figure 2B, 4B, Figure 5I as examples).
- Figure 1A needs a scale for understanding the significance of bubble size.
- Figure 2A displays overlapping flow cytometry gates. Please adjust these so that events are not being counted within both populations.
- Figure 6A western blotting analysis would benefit from quantitative analysis and/or specification of repeat blots within the supplemental figures.
- Figure 6C only has two sizes listed for P-values but there are three sizes within the plot. Please clarify what the smallest dot size represents.
- Please include the clones and manufacturer of the antibodies used in the naïve CD8 T cell isolation (lines 599 - 600)

Reviewer #2 (expert in T cell memory):

This is a very well-executed study showing a striking failure of T cell population expansion upon

TCR triggering in Cul4b deficient hosts. It establishes that the central driver of T cell proliferation, c-Myc, increases expression of Cul4b, which then localizes to the chromatin to limit the accumulation of proteins implicated in replication stress. While the authors have shown some of these effects already in the context of CD4 T cell activation, sufficient novelty persists due to the extensive *in vivo* data (infection models, effects in viral titres, mouse survival etc.) presented and the link to c-Myc that is established for the first time.

I would like the authors to address three more points to provide a fully rounded manuscript:

- 1) Does the observed replication stress in Cul4b deficient CD8 T cells limit T cell expansion upon (e.g. LCMV infection) via restriction of proliferation and/or increased induction of apoptosis?
- 2) Is PD-1 upregulation and T cell exhaustion a direct consequence of replication stress or induced due to limited population expansion and the resulting failure to clear LCMV-Arm infection. This could be answered by investigating PD-1 expression and cytokine production in Cul4b deficient P14 T cells transferred to WT hosts and subsequently infected with LCMV-Arm. In this setting virus will be controlled by the endogenous T cells excluding viral persistence as a potential culprit.
- 3) The manuscript should be shortened and sharpened. It feels a bit encyclopaedic and at times loses focus with respect to the authors' key messages.

Reviewer #3 (expert in cell cycle regulation):

This study identifies a role for Cul4b downstream of c-Myc as a regulator of cell proliferation in CD8+ T cells. The study shows that Cul4b is important for activation-induced and homeostatic proliferation of CD8+ T cells. Strikingly, condition deletion of Cul4b in CD4+ expressing cells results in defective memory responses to LCMV *in vivo*. Mechanistically, Cul4b deletion results in increased DNA damage and genome instability in proliferating CD8+ T cells, leading to arrest/delays in G2&M phases of the cell cycle and presumably cell death. It remains unclear why Cul4b increases DNA damage in CD8+ T cells, although a role for Cyclin E2 and p21 overexpression are implicated. Notably, p21 is highly induced following TCR activation (e.g., <http://www.immpres.co.uk/>), but the absolute levels are important for the function of p21 as a CDK inhibitor protein, and Cul4bcKO appears to produce even higher levels of p21. These are important findings that will be of interest to the field.

Major points:

The manuscript requires a significant re-write as there are grammatical issues throughout that make interpretation of the study challenging.

Does Cul4bcKO affect cell survival?

Is there a difference in cell size in the activated MyccKO and Cul4bcKO CD8+ T cells compared to WT?

Is there re-replication in the activated Cul4bcKO CD8+ T cells?

RESPONSE TO REVIEWERS' COMMENTS

Reviewer #1 (expert in T cell memory):

This manuscript builds upon previous work from the Oliver lab investigating the role of Cul4b regulation of DNA damage response in CD4 T cells. Here the authors identify a role for Cul4b in CD8 T cell proliferation, effector response, and memory formation. Through transcriptomic, proteomic, and cellular immunologic methods, the authors identify a regulatory network where c-Myc upregulates Cul4b in T cells to mitigate activation-induced DNA damage and cell cycle entrance through regulation of p21 and Cyclin E2.

This manuscript provides insight into the molecular mechanisms of managing T cell replicative stress. While this manuscript provides new insight into the interactions between c-Myc, Cul4b, p21, and Cyclin E2, there are a few concerns regarding the model and molecular mechanisms of these interactions.

Major Comments:

- **Data supporting normal thymic development of the Cul4b-cKO animals would be important to demonstrate that the Cul4b naïve T cells are comparable to WT cells prior to activation. Enumeration of thymic developmental stages as well as repertoire analysis would be beneficial. In the Dar AA 2021 PLoS Biology publication, there were data to suggest decreased CD8 T cells in the thymus and spleen of Cul4b-cKO mice which may impact the interpretations of the peripheral T cell responses examined within this manuscript. An inducible knock-out system may complement or improve the design of these experiments to determine the contribution of Cul4b in peripheral T cell responses and remove any confounding variables from the proliferative burst during thymic development.**

Answer: This is an important point. Cul4, substrate receptor DCAF1 has been shown to interact with RAG and regulate its expression hence influence the TCR rearrangement (Kassmeier, Mondal et al. 2012) (Schabla, Perry et al. 2018) . In this model, Cul4b is very likely deleted after CD4 is expressed, and thus after TCR rearrangement. Nonetheless, it is important for us show that the TCR repertoire is not impacted. In our previous study we reported that there was a marginal decrease in CD8⁺ T cell percentages in the thymus and periphery however these differences were not seen when we assessed cell numbers. To assess this in greater detail, we performed flow cytometric analysis of the TCR V β repertoire in thymic T cells from control and Cul4b^{cKO} mice. The usage of several V β chains in Cul4b deficient T cells was compared to the control cells. We did not observe any significant impact of TCR V β chain usage for any chains tested. TCR repertoire data from thymic SP CD8⁺ T cells is now shown in the supplementary figure 2. The similarities of TCR V β repertoire between these control mice and Cul4^{cKO} mice indicates that Cul4b deletion at the DP stage did not change the TCR repertoire and hence is unlikely to impact the outcome of current study. The changes made to the text are highlighted in yellow at line 211-213.

- The authors investigate the impairment of the DNA damage response in Cul4b-cKO cells but, it would be helpful to use an additional functional readout like a Comet assay that was used in the Dar AA et al paper. The addition of this assay would complement other data in Figure 7 of this manuscript to strengthen the evidence of increased DNA damage in Cul4b-cKO CD8 T cells.

Answer: As suggested by the reviewer, we performed a comet assay to assess the extent of the DNA damage in activated control and Cul4b^{CKO} CD8⁺ T cells after Camptothecin treatment. The comet assay revealed that Cul4b^{CKO} CD8⁺ T cells showed more DNA damage than their control counterparts. The analyzed data is now included in the figure 7d,e and necessary changes are made in the text and highlighted in yellow at line 398-401, 600-610. Overall, our data provides strong evidence that CD8⁺ T cells lacking Cul4b have increased DNA damage.

Minor Comments:

- The authors propose an interactive model between c-Myc, Cul4b, p21, and Cyclin E2. It would be beneficial to provide a summary diagram that highlights these interactions in the WT and KO conditions.

Answer: As requested by the reviewer we have included a pathway diagram to summarize our findings. The pathway diagram is included as supplementary figure 9i

- Additional citations within the introduction and results would be helpful in lines 57, 73, 77, 107.

Answer: As suggested, we have included the relevant citations

- A description of the c-Myc conditional knock-out mouse model is missing from the methods section.

Answer: We would like to clarify that we did not maintain the c-Myc mice colony, the data on c-Myc was analyzed from the publicly available datasets. For each dataset the relevant source was cited in the results and method section. For example, line 119, 715, 742, 760,767. However, at the places where we missed citing the source, we have cited it in the revised manuscript for example line 391.

- In line 306, the marker is listed as 'CXC3CR1' instead of 'CX3CR1'.

Answer: Thanks for pointing out this we corrected the typo

- In line 322, the authors state that the number of P14 cells is decreased but this should state frequency since enumeration of cells was not shown for this data.

Answer: We thank reviewer for noticing this. As the control and Cul4b^{CKO} P14 cells were maintained in the same recipient mice, the decrease in percentages would be inevitably reflected at numbers. However, for clarity we replaced numbers with percentages at line 306 in revised manuscript.

- The figures would benefit from using open/closed circles or another strategy to distinguish between experimental groups in graphs. When printed in black and white, it is very difficult to tell which group is which in the figures (see Figure 2B, 4B, Figure 5I as examples).

Answer: We used the strategy of representing data points with open circles. The control group is represented in black circle while Cul4b^{CKO} group in burgundy red. For the sake of convenience, we have modified the line graphs for example figure 2b, 3b, 4g, h while the bar graphs have been labelled wherever labeling was missing for example 5b, 5d, 5f, 2k, etc

- **Figure 1A needs a scale for understanding the significance of bubble size.**

Answer: The size or area of the displayed circles is proportional to the number of genes assigned to the term. The statement is included in the figure legend and also figures have been updated in the revised manuscript

- **Figure 2A displays overlapping flow cytometry gates. Please adjust these so that events are not being counted within both populations.**

Answer: As suggested, we have adjusted the gates

- **Figure 6A western blotting analysis would benefit from quantitative analysis and/or specification of repeat blots within the supplemental figures.**

Answer: The replicates of the experiment will be included in the source data

- **Figure 6C only has two sizes listed for P-values but there are three sizes within the plot. Please clarify what the smallest dot size represents.**

Answer: The p values corresponding to the smallest circle has been included in the figure

- **Please include the clones and manufacturer of the antibodies used in the naive CD8 T cell isolation (lines 599- 600)**

Answer: The antibodies are provided in the kit (MACS Miltenyi Kit) as indicated in the line 583. The manufacturer does not disclose the clone of the antibodies.

Reviewer #2 (expert in T cell memory):

This is a very well-executed study showing a striking failure of T cell population expansion upon TCR triggering in Cul4b deficient hosts. It establishes that the central driver of T cell proliferation, c-Myc, increases expression of Cul4b, which then localizes to the chromatin to limit the accumulation of proteins implicated in replication stress. While the authors have shown some of these effects already in the context of CD4 T cell activation, sufficient novelty persists due to the extensive in vivo data (infection models, effects in viral titres, mouse survival etc.) presented and the link to c-Myc that is established for the first time.

I would like the authors to address three more points to provide a fully rounded manuscript:

1) Does the observed replication stress in Cul4b deficient CD8 T cells limit T cell expansion upon (e.g. LCMV infection) via restriction of proliferation and/or increased induction of apoptosis?

Answer: This is an important clarification. As we have shown that Cul4b-deficient CD8⁺ T cells proliferated and expanded less than control cells as evidenced in Figure 2A-D. The reduced expansion of Cul4b-deficient CD8⁺ T cells could also be an indication of their poor survival rate. To test this, we stimulated cells under in vitro culture conditions and analyzed cell death using Annexin V staining. We found significantly higher percentages of Cul4b-deficient CD8⁺ T cells undergoing apoptosis (fig 7k and supplementary fig. 9f). However, when cell proliferation was blocked using rapamycin the control and Cul4b-deficient CD8⁺ T did not show any difference in cell death (supplementary fig. 9g, h). This indicates that in absence of Cul4b, CD8⁺ T cells are not able to circumvent the replication stress hence proliferate less and also undergo apoptotic death.

2) Is PD-1 upregulation and T cell exhaustion a direct consequence of replication stress or induced due to limited population expansion and the resulting failure to clear LCMV-Arm infection. This could be answered by investigating PD-1 expression and cytokine production in Cul4b deficient P14 T cells transferred to WT hosts and subsequently infected with LCMV-Arm. In this setting virus will be controlled by the endogenous T cells excluding viral persistence as a potential culprit.

Answer: We thank reviewer for making this point. As suggested, we transferred the control and Cul4b^{CKO} P14 T cells into a congenically distinct WT mice. The recipient cells were analyzed for the PD-1 expression at day 30 post infection. Neither donor cells (control and Cul4b^{CKO} P14 T) or recipient cells expressed PD-1. This clearly indicates that the increased PD-1 expression found on Cul4b-deficient CD8⁺ T cells was due to their inability to clear the virus. The data is included in the supplementary figure 6J and this is now clarified in the text. The ability to produce cytokines is dependent on Cul4b in a cell intrinsic fashion as shown in figure 2I and Supplementary I,j,k).

3) The manuscript should be shortened and sharpened. It feels a bit encyclopaedic and at times loses focus with respect to the authors' key messages.

Answer: We have attempted to reduce the text to make the manuscript more concise.

Reviewer #3 (expert in cell cycle regulation):

This study identifies a role for Cul4b downstream of c-Myc as a regulator of cell proliferation in CD8+ T cells. The study shows that Cul4b is important for activation-induced and homeostatic proliferation of CD8+ T cells. Strikingly, condition deletion of Cul4b in CD4+ expressing cells results in defective memory responses to LCMV in vivo. Mechanistically, Cul4b deletion results in increased DNA damage and genome instability in proliferating CD8+ T cells, leading to arrest/delays in G2&M phases of the cell cycle and presumably cell death. It remains unclear why Cul4b increases DNA damage in CD8+ T cells, although a role for Cyclin E2 and p21 overexpression are implicated. Notably, p21 is highly induced following TCR activation (e.g., <http://www.immpres.co.uk/>), but the absolute levels are important for the function of p21 as a CDK inhibitor protein, and Cul4b^{ckO} appears to produce even higher levels of p21. These are important findings that will be of interest to the field.

Major points:

The manuscript requires a significant re-write as there are grammatical issues throughout that make interpretation of the study challenging.

We have proof read the manuscript for grammatical issues-additional changes will likely be made during the editorial process

Does Cul4b^{ckO} affect cell survival?

Answer: We would like to thank the reviewer for the comment. To confirm this, we stimulated both control and Cul4b deleted CD8⁺ T cells in vitro, allowed them to proliferate for three days, and then used Annexin V staining to examine cell death. We found significantly higher percentages of Cul4b-deficient CD8⁺ T cells undergoing apoptosis (Figure 7k and Supplementary 9f). The data indicates that Cul4b-deficient CD8⁺ T cells survive less.

Is there a difference in cell size in the activated Myc^{ckO} and Cul4b^{ckO} CD8+ T cells compared to WT?

Answer: The difference in the cell size of MYC^{ckO} compared to WT cells is already established (Marchingo, Sinclair et al. 2020). Myc-deficient T cells do not substantially increase cell size or proliferate in response to immune activation with anti-CD3/anti-CD28 agonist antibodies. However, we did not find any difference in the size of surviving Cul4b^{ckO} CD8⁺ T cells compared to WT. Forward scatter area (FSC-A) of anti-CD3 + anti-CD28 (TCR) activated control and Cul4b deficient CD8⁺ T cells is shown in the Supplementary Fig.1i. This is stated in the revised manuscript line 166-167

Is there re-replication in the activated Cul4b^{CKO} CD8⁺ T cells?

Answer: To check whether Cul4b deficiency induces re-replication, the DNA content of control and Cul4b^{CKO} CD8⁺ T cells was analyzed. Cul4b-deficient CD8⁺ T cells showed significantly increased numbers of cells with DNA >4N at day 2 after activation. These data support re-replication (Fig.7j and Supplementary Fig. 9e). At later time point, day 3, this was associated with increased cell death as shown in Figure 7k and Supplementary Fig.9f). These results suggest that re-replication induced by a lack of Cul4b ultimately results in apoptosis in CD8⁺ T cells

Kassmeier, M. D., K. Mondal, V. L. Palmer, P. Raval, S. Kumar, G. A. Perry, D. K. Anderson, P. Ciborowski, S. Jackson, Y. Xiong and P. C. Swanson (2012). "VprBP binds full-length RAG1 and is required for B-cell development and V(D)J recombination fidelity." *EMBO J* **31**(4): 945-958.

Marchingo, J. M., L. V. Sinclair, A. J. Howden and D. A. Cantrell (2020). "Quantitative analysis of how Myc controls T cell proteomes and metabolic pathways during T cell activation." *Elife* **9**.

Schabla, N. M., G. A. Perry, V. L. Palmer and P. C. Swanson (2018). "VprBP (DCAF1) Regulates RAG1 Expression Independently of Dicer by Mediating RAG1 Degradation." *J Immunol* **201**(3): 930-939.

REVIEWER COMMENTS

Reviewer #1 (expert in T cell memory):

The revised manuscript has addressed the major concerns, is improved, and acceptable for publication.

Reviewer #2 (expert in T cell memory):

The authors have sufficiently addressed my concerns.

Reviewer #3 (expert in cell cycle regulation):

The authors have submitted a revised manuscript including additional data that mostly addresses my concerns. However, there are several instances where the authors unnecessarily overstate the conclusions that can be substantiated from the data. This can be fixed by changes in wording.

1. As noted in my original review, it is unclear why Cul4b deletion increases DNA damage in CD8+ T cells. The authors convincingly show that p21 and Cyclin E play roles, but there is no compelling evidence to show that high p21 levels cause DNA damage in CD8+ T cells (only citing ref 36, an earlier paper that showed that high p21 levels can induce DNA damage in another cellular context). Therefore, I am concerned by the interpretation of the results, e.g. including line 418, "p21 induced replication stress in Cul4b-deficient CD8+ T is likely the source of DNA damage.", and discussion starting on line 480. Indeed, the increase in p21 mRNA suggests that p21 might be a consequence of DNA damage caused by Cul4b loss (not exclusively the cause as the authors currently argue). The Discussion should be reworded to reflect this accurately.

2. Line 44: mechanisms regulating CD8+ T cell cycle are well studied -- is this really true? Perhaps there should be a reference cited.

3. Line 388: "Elevated expression of p21 in a p53 neutral environment resulted in genomic instability, as proteomic analysis identified deregulation of the replication licensing machinery as major cause of genome instability." The authors do not show genome instability. DNA damage does not equate to genome instability because damage in principle could be repaired. I recommend rephrasing here.

4. Line 402: "This led us to examine what was causing Cul4b deficient CD8+ T cells to enter into S-phase in a p21 high state." This was not shown.

5. Line 417: "p21 induced replication stress" cause or consequence?

That aside, I think the study is very strong and worth publication if the concerns above related to the discussion of the data are addressed.

RESPONSE TO REVIEWERS' COMMENTS

Reviewer #1 (expert in T cell memory):

The revised manuscript has addressed the major concerns, is improved, and acceptable for publication.

Reviewer #2 (expert in T cell memory):

The authors have sufficiently addressed my concerns.

Reviewer #3 (expert in cell cycle regulation):

The authors have submitted a revised manuscript including additional data that mostly addresses my concerns. However, there are several instances where the authors unnecessarily overstate the conclusions that can be substantiated from the data. This can be fixed by changes in wording.

1. As noted in my original review, it is unclear why Cul4b deletion increases DNA damage in CD8+ T cells. The authors convincingly show that p21 and Cyclin E play roles, but there is no compelling evidence to show that high p21 levels cause DNA damage in CD8+ T cells (only citing ref 36, an earlier paper that showed that high p21 levels can induce DNA damage in another cellular context). Therefore, I am concerned by the interpretation of the results, e.g. including line 418, "p21 induced replication stress in Cul4b-deficient CD8+ T is likely the source of DNA damage.", and discussion starting on line 480. Indeed, the increase in p21 mRNA suggests that p21 might be a consequence of DNA damage caused by Cul4b loss (not exclusively the cause as the authors currently argue). The Discussion should be reworded to reflect this accurately.

Answer: We appreciate the point that reviewer has raised.

We have edited the statements on lines 418 to say "Increased levels of Cyclin E2 and p21 are likely to increase replication stress in Cul4b-deficient CD8+ T and this may be the source of DNA damage." Additionally, we have modified the wording in the discussion to more accurately reflect the data. Specifically, we concluded the paragraph starting on line 496 with "our data support a model in which Cul4b restrains replication stress likely by controlling levels of Cyclin E2 and p21".

2. Line 44: mechanisms regulating CD8+ T cell cycle are well studied -- is this really true? Perhaps there should be a reference cited.

Answer: We have rephrased the statement at line 44 to say "Mechanisms regulating CD8+ T cell cycle and DDR pathways during the generation of effector and/or memory CD8+ T cells remain poorly defined."

3. Line 388: "Elevated expression of p21 in a p53 neutral environment resulted in genomic instability, as proteomic analysis identified deregulation of the replication licensing machinery as major cause of genome instability." The authors do not show genome instability. DNA damage does not equate to genome instability because damage in principle could be repaired. I recommend rephrasing here.

Answer: We have modified the statement on line 388. It now reads "Elevated expression of p21 in a p53 neutral environment has been shown to cause genomic instability through deregulation of the replication licensing machinery".

4. Line 402: "This led us to examine what was causing Cul4b deficient CD8+ T cells to enter into S-phase in a p21 high state." This was not shown.

Answer: We modified the statement to read as follows "This led us to examine what might be causing Cul4b deficient CD8+ T cells to enter into S-phase when p21 levels were measurably elevated."

5. Line 417: "p21 induced replication stress" cause or consequence?

Answer: This was modified as stated in comment 1 above.

That aside, I think the study is very strong and worth publication if the concerns above related to the discussion of the data are addressed.